# NerveNet: Learning Structured Policy with Graph Neural Networks

**Tingwu Wang**\*, **Renjie Liao**\*, **Jimmy Ba & Sanja Fidler**
Department of Computer Science
University of Toronto
Vector Institute
{tingwuwang,rjliao}@cs.toronto.edu,
jimmy@psi.toronto.edu, fidler@cs.toronto.edu

## Abstract

We address the problem of learning structured policies for continuous control. In traditional reinforcement learning, policies of agents are learned by multi-layer perceptrons (MLPs) which take the concatenation of all observations from the environment as input for predicting actions. In this work, we propose NerveNet to explicitly model the structure of an agent, which naturally takes the form of a graph. Specifically, serving as the agent's policy network, NerveNet first propagates information over the structure of the agent and then predict actions for different parts of the agent. In the experiments, we first show that our NerveNet is comparable to state-of-the-art methods on standard MuJoCo environments. We further propose our customized reinforcement learning environments for benchmarking two types of structure transfer learning tasks, i.e., *size* and *disability transfer*, as well as *multi-task learning*. We demonstrate that policies learned by NerveNet are significantly more transferable and generalizable than policies learned by other models and are able to transfer even in a zero-shot setting.

## 1 Introduction

Deep reinforcement learning (RL) has received increasing attention over the past few years, with the recent success of applications such as playing Atari Games, Mnih et al. (2015), and Go, (Silver et al., 2016; 2017). Significant advances have also been made in robotics using the latest RL techniques, e.g., Levine et al. (2016); Gu et al. (2017).

Many RL problems feature agents with multiple dependent controllers. For example, humanoid robots consist of multiple physically linked joints. Action to be taken by each joint or the body should thus not only depend on its own observations but also on actions of other joints.

Previous approaches in RL typically use MLP to learn the agent's policy. In particular, MLP takes the concatenation of observations from the environment as input, which may be measurements like positions, velocities of body and joints in the current time instance. The MLP policy then predicts actions to be taken by every joint and body. Thus the task of the MLP policy is to discover the latent relationships between observations. This typically leads to longer training times, requiring more exposure of the agent to the environment. In our work, we aim to exploit the body structure of an agent, and physical dependencies that naturally exist in such agents.

We rely on the fact that bodies of most robots and animals have a discrete graph structure. Nodes of the graph may represent the joints, and edges represent the (physical) dependencies between them. In particular, we define the agent's policy using a Graph Neural Network, Scarselli et al. (2009), which is a neural network that operates over graph structures. We refer to our model as NerveNet due to the resemblance of the neural nervous system to a graph. NerveNet propagates information between different parts of the body based on the underlying graph structure before outputting the action for each part. By doing so, NerveNet can leverage the structure information encoded by the agent's body which is advantageous in learning the correct inductive bias, and thus is less prone to

---

\*Two authors contribute equally.

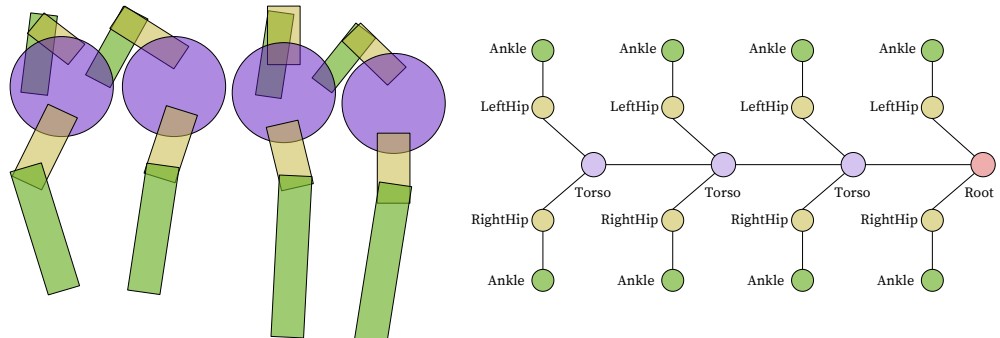

Figure 1: Visualization of the graph structure of `CentipedeEight` in our environment. We use this agent for testing the ability of transfer learning of our model. Since for this agent, each `body` node is paired with at least one `joint` node, we omit the `body` nodes and fill up the position with the corresponding `joint` nodes. By omitting the `body` nodes, a more compact graph is constructed, the details of which are illustrated in the experimental section.

overfitting. Moreover, NerveNet is naturally suitable for structure transferring tasks as most of the model weights are shared across the nodes and edges, respectively.

We first evaluate our NerveNet on standard RL benchmarks such as the OpenAI Gym, Brockman et al. (2016) which stem from MuJoCo. We show that our model achieves comparable results to state-of-the-art MLP based methods. To verify our claim regarding the structure transfer, we further introduce our customized RL environments which are based on the ones of Gym. Two types of structure transfer tasks are designed, *size transfer* and *disability transfer*. In particular, *size transfer* focuses on the scenario in which policies are learned for small-sized agents (simpler body structure) and applied directly to large-sized agents which are composed by some repetitive components shared with the small-sized agent. Secondly, *disability transfer* investigates scenarios in which policies are learned for one agent and applied to the same agent with some components disabled. Our experiments demonstrate that for structure transfer tasks our NerveNet is significantly better than all other competitors, and can even achieve zero-shot learning for some agents. For the multi-task learning tasks, NerveNet is also able to learn policies that are more robust with better efficiency.

The main contribution of this paper is the following: We explore the problem of learning transferable and generalized features by incorporating a prior on the structure via graph neural networks. NerveNet permits powerful transfer learning from one structure to another, which goes well beyond the ability of previous models. NerveNet is also more robust and has more potential in performing multi-task learning. The demo and code for this project are released, under the project page of http://www.cs.toronto.edu/~tingwuwang/nervenet.html.

## 2 NERVENET

In this section, we first introduce the notation. We then explain how to construct the graph for each of our agents, followed by the description of the NerveNet. Finally, we describe the learning algorithm for our model.

We formulate the locomotion control problems as an infinite-horizon discounted Markov decision process (MDP). To fully describe the MDP for continuous control problems which include locomotion control, we define the state space or observation space as $\mathcal{S}$ and action space as $\mathcal{A}$. To interact with the environments, the agent generates its stochastic policy $\pi_\theta(a^\tau|s^\tau)$ based on the current state $s^\tau \in \mathcal{S}$, where $a^\tau \in \mathcal{A}$ is the action and $\theta$ are the parameters of the policy function. The environment on the other hand, produces a reward $r(s^\tau, a^\tau)$ for the agent, and the agent's objective is to find a policy that maximizes the expected reward.

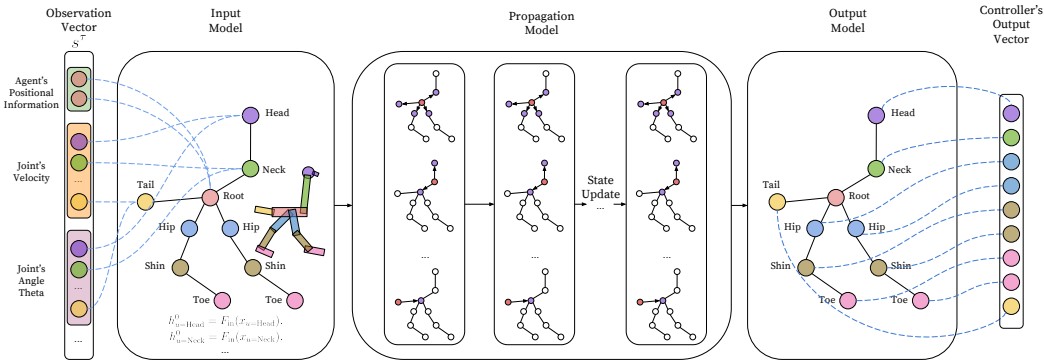

Figure 2: In this figure, we use `Walker-Ostrich` as an example of NerveNet. In the input model, for each node, NerveNet fetches the corresponding elements from the observation vector. NerveNet then computes the messages between neighbors in the graph, and updates the hidden state of each node. This process is repeated for a certain number of propagation steps. In the output model, the policy is produced by collecting the output from each controller.

## 2.1 GRAPH CONSTRUCTION

In real life, skeletons of most robots and animals have a discrete graph structure, and are most often trees. Simulators such as the MuJoCo engine by Todorov et al. (2012), organize the agents using an XML-based kinematic tree. In our experiments, we will use the tree graphs as per MuJoCo. Note that our model can be directly applied to arbitrary graphs. In particular, we assume two types of nodes in our tree: `body` and `joint`. The `body` nodes are abstract nodes used to construct the kinematic tree via nesting, which is similar to the coordinate frame system used in robotics, Spong et al. (2006). The `joint` node represents the degrees of freedom of motion between the two `body` nodes. Take a simple humanoid as an example; the `body` nodes `Thigh` and `Shin` are connected via the `Knee`, where `Knee` is a hinge `joint`. We further add a `root` node which observes additional information about the agent. For example, in the `Reacher` agent in MuJoCo, the `root` node has access to the target position of the agent. We build edges to form a tree graph. Fig. 1 illustrates the graph structure of an example agent, `CentipedeEight`. We sketch the agent and its corresponding graph in the left and right part of the figure, respectively. Note that for better visualization, we omit the `joint` nodes and use edges to represent the physical connections of `joint` nodes. Different elements of the agent are parsed into nodes with different colors. Further details are provided in the experimental section.

## 2.2 NERVENET AS POLICY NETWORK

We now turn to NerveNet which parametrizes the policy with a Graph Neural Network. Before delving into details, we first introduce our notation. We then specify the input model which helps to initialize the hidden state of each node. We further introduce the propagation model that updates these hidden states. Finally, we describe the output model.

We denote the graph structure of the agent as $G = (V, E)$ where $V$ and $E$ are the sets of nodes and edges, respectively. We focus on the directed graphs as the undirected case can be easily addressed by splitting one undirected edge into two directed edges. We denote the out-going neighborhood of node $u$ as $\mathcal{N}_{out}(u)$ which contains all endpoints $v$ with $(u, v)$ being an edge in the graph. Similarly, we denote the in-coming neighborhood of node $u$ as $\mathcal{N}_{in}(u)$. Every node $u$ has an associated node type $p_u \in \{1, 2, \ldots, P\}$, which in our case corresponds to `body`, `joint` and `root`. We also associate each edge $(u, v)$ with an edge type $c_{(u,v)} \in \{1, 2, \ldots, C\}$. Node type can help in capturing different importances across nodes. Edge type can be used to describe different relationships between nodes, and thus propagate information between them differently. One can also add more than one edge type to the same edge which results in a multi-graph. We stick to simple graphs for simplicity. One interesting fact is that we have two notions of "time" in our model. One is the time step in the environment which is the typical time coordinate for RL problems. The other corresponds

to the internal propagation step of NerveNet. These two coordinates work as follows. At each time step of the environment, NerveNet receives observation from the environment and performs a few internal propagation steps in order to decide on the action to be taken by each node. To avoid confusion, throughout this paper, we use $\tau$ to describe the time step in the environment and $t$ for the propagation step.

### 2.2.1 INPUT MODEL

For each time step $\tau$ in the environment, the agent receives an observation $s^\tau \in \mathcal{S}$. The observation vector $s^\tau$ is the concatenation of observations of each node. We denote the elements of observation vector $s^\tau$ corresponding to node $u$ with $x_u$. From now on, we drop the time step in the environment to derive the model for simplicity. The observation vector goes through an input network to obtain a fixed-size state vector as follows:

$$h_u^0 = F_{\text{in}}(x_u), \tag{1}$$

where the subscript and superscript denote the node index and propagation step, respectively. Here, $F_{\text{in}}$ may be a MLP and $h_u^0$ is the state vector of node $u$ at propagation step 0. Note that we may need to pad zeros to the observation vectors if different nodes have observations of different sizes.

### 2.2.2 PROPAGATION MODEL

We now describe the propagation model of our NerveNet which mimics a synchronous message passing system studied in distributed computing, Attiya & Welch (2004). We will show how the state vector of each node is updated from one propagation step to the next. This update process is recurrently applied during the whole propagation. We leave the details to the appendix.

**Message Computation** In particular, at propagation step $t$, for every node $u$, we have access to a state vector $h_u^t$. For every edge $(u, v) \in \mathcal{N}_{out}(u)$, node $u$ computes a message vector as below,

$$m_{(u,v)}^t = M_{c_{(u,v)}}(h_u^t), \tag{2}$$

where $M_{c_{(u,v)}}$ is the message function which may be an identity mapping or a MLP. Note that the subscript $c_{(u,v)}$ indicates that edges of the same edge type share the same instance of the message function. For example, the second `torso` in Fig. 1 sends a message to the first and third `torso`, as well as the `LeftHip` and `RightHip`.

**Message Aggregation** Once every node finishes computing messages, we aggregate messages sent from all in-coming neighbors of each node. Specifically, for every node $u$, we perform the following aggregation:

$$\bar{m}_u^t = A(\{h_v^t | v \in \mathcal{N}_{in}(u)\}), \tag{3}$$

where $A$ is the aggregation function which may be a summation, average or max-pooling function. Here, $\bar{m}_u^t$ is the aggregated message vector which contains the information sent from the node's neighborhood.

**States Update** We now update every node's state vector based on both the aggregated message and its current state vector. In particular, for every node $u$, we perform the following update:

$$h_u^{t+1} = U_{p_u}(h_u^t, \bar{m}_u^t), \tag{4}$$

where $U$ is the update function which may be a gated recurrent unit (GRU), a long short term memory (LSTM) unit or a MLP. From the subscript $p_u$ of $U$, we can see that nodes of the same node type share the same instance of the update function. The above propagation model is then recurrently applied for a fixed number of time steps $T$ to get the final state vectors of all nodes, i.e., $\{h_u^T | u \in V\}$.

### 2.2.3 OUTPUT MODEL

In RL, agents typically use a MLP policy, where the network outputs the mean of the Gaussian distribution for each of the actions, while the standard deviation is a trainable vector, Schulman et al. (2017). In our output model, we also treat standard deviation in the same way.

However, instead of predicting the action distribution of all nodes by a single network, we make predictions for each individual node. We denote the set of nodes which are assigned controllers for the actuators as $\mathcal{O}$. For each such node, a MLP takes its final state vectors $h_{u\in\mathcal{O}}^T$ as input and produces the mean of the action of the Gaussian policy for the corresponding actuator. For each output node $u \in \mathcal{O}$, we define its output type as $q_u$. Different sharing schemes are available for the instance of MLP $O_{q_u}$, for example, we can force the nodes with similar physical structure to share the instance of MLP. For example, in Fig. 1, two `LeftHip` nodes have a shared controller. Therefore, we have the following output model:

$$\mu_{u\in\mathcal{O}} = O_{q_u}(h_u^T), \tag{5}$$

where $\mu_{u\in\mathcal{O}}$ is the mean value for action applied on each actuator. In practice, we found that we can force controllers of different output types to share one unified controller, while not hurting the performance. By integrating the produced Gaussian policy for each action, the probability density of the stochastic policy is calculated as

$$\pi_\theta(a^\tau|s^\tau) = \prod_{u\in\mathcal{O}} \pi_{\theta,u}(a_u^\tau|s^\tau) = \prod_{u\in\mathcal{O}} \frac{1}{\sqrt{2\pi\sigma_u^2}} e^{(a_u^\tau - \mu_u)^2/(2\sigma_u^2)}, \tag{6}$$

where $a^\tau \in \mathcal{A}$ is the output action, and $\sigma_u$ is the variable standard deviation for each action. Here, $\theta$ represents the parameters of the policy function.

## 2.3 Learning Algorithm

To interact with the environments, the agent generates its stochastic policy $\pi_\theta(a^\tau|s^\tau)$ after several propagation steps. The environment on the other hand, produces a reward $r(s^\tau, a^\tau)$ for the agent, and transits to the next state with transition probability $P(s^{\tau+1}|s^\tau)$. The target of the agent is to maximize its cumulative return

$$J(\theta) = \mathbb{E}_\pi \left[ \sum_{\tau=0}^\infty \gamma^\tau r(s^\tau, a^\tau) \right]. \tag{7}$$

To optimize the expected reward, we use the proximal policy optimization (PPO) by Schulman et al. (2017). In PPO, the agents alternate between sampling trajectories with the latest policy and performing optimization on surrogate objective using the sampled trajectories. The algorithm tries to keep the KL-divergence of the new policy and the old policy within the trust region. To achieve that, PPO clips the probability ratio and adds an KL-divergence penalty term to the loss. The likelihood ratio is defined as $r^\tau(\theta; \theta_{old}) = \pi_\theta(a^\tau|s^\tau)/\pi_{\theta_{old}}(a^\tau|s^\tau)$. Following the notation and the algorithm of PPO, our NerveNet tries to minimize the summation of the original loss in Eq. (7), KL-penalty and the value function loss which is defined as:

$$
\begin{aligned}
\tilde{J}(\theta) =& J(\theta) - \beta L_{KL}(\theta) - \alpha L_V(\theta) \\
=& \mathbb{E}_{\pi_\theta} \left[ \sum_{\tau=0}^\infty \min\left( \hat{A}^\tau r^\tau(\theta), \hat{A}^\tau \operatorname{clip}\left( r^\tau(\theta), 1-\epsilon, 1+\epsilon \right) \right) \right] \\
& - \beta \mathbb{E}_{\pi_\theta} \left[ \sum_{\tau=0}^\infty \operatorname{KL}\left[ \pi_\theta(:|s^\tau)|\pi_{\theta_{old}}(:|s^\tau) \right] \right] - \alpha \mathbb{E}_{\pi_\theta} \left[ \sum_{\tau=0}^\infty \left( V_\theta(s^\tau) - V(s^\tau)^{\text{target}} \right)^2 \right], \quad (8)
\end{aligned}
$$

where $\hat{A}_t$ the generalized advantage estimation (GAE) calculated using algorithm from Schulman et al. (2015b), and $\epsilon$ is the clip value, which we choose to be $0.2$. Here, $\beta$ is a dynamical coefficient adjusted to keep the KL-divergence constraints, and $\alpha$ is used to balance the value loss. Note that in Eq. (8), $V(s_t)^{\text{target}}$ is the target state value in accordance with the GAE method. To optimize the $\tilde{J}(\theta)$, PPO make use of the policy gradient in Sutton et al. (2000) to do first-order gradient descent optimization.

**Value Network**   To produce the state value $V_\theta(s^\tau)$ for given observation $s^\tau$, we have several alternatives: (1) using one GNN as the policy network and using one MLP as the value network (NerveNet-MLP); (2) using one GNN as policy network and using another GNN as value network (NerveNet-2) (without sharing the parameters of the two GNNs); (3) using one GNN as both policy network and value network (NerveNet-1). The GNN for value network is very similar to the GNN for policy network. The output for value GNN is a scalar instead of a vector of mean action. We will compare these variants in the experimental section.

## 3 RELATED WORK

**Reinforcement Learning**   Reinforcement learning (RL) has recently achieved huge success in a variety of applications. Powered by the progress of deep neural networks, Krizhevsky et al. (2012), agents are now able to successfully play Atari Games and beat the world's best (human) players in the game of Go (Mnih et al., 2015; Silver et al., 2016; 2017). Based on simulation engines like MuJoCo, Todorov et al. (2012), numerous algorithms have been proposed to train agents also in continuous control problems (Schulman et al., 2017; 2015a; Heess et al., 2017; Metz et al., 2017).

**Structure in RL**   Most approaches that exploit priors on structure of the problem fall in the domain of hierarchical RL, (Kulkarni et al., 2016; Vezhnevets et al., 2017), which mainly focus on modeling intrinsic motivation of agents. In Hausknecht & Stone (2015), the authors extend the deep RL algorithms to MDPS with parameterized action space by exploiting the structure of action space and bounding the action space gradients. Graphs have been used in RL problems prior to our work. In Metzen (2013); Mabu et al. (2007); Shoeleh & Asadpour (2017); Mahadevan & Maggioni (2007), the authors use graphs to learn a representation of the environment. However, these methods are limited to problems with simple dynamical models like for example the task of 2d-navigation, and thus these problems are usually solved via model-based RL. However, for complex multi-joint agents, learning the dynamical model as well as predicting the transition of states is time consuming and biased. For problems of training model-free multi-joint agents in complex physical environments, relatively little attention has been devoted to modeling the physical structure of the agents.

**Graph Neural Networks**   There have been many efforts to generalize neural networks to graph-structured data. One line of work is based on convolutional neural networks (CNNs). In (Bruna et al., 2014; Defferrard et al., 2016; Kipf & Welling, 2017), CNNs are employed in the spectral domain relying on the graph Laplacian matrix. (Goller & Kuchler, 1996; Duvenaud et al., 2015) used hash functions in order to apply CNNs to graphs.

Another popular direction is based on recurrent neural networks (RNNs) (Goller & Kuchler, 1996; Gori et al., 2005; Scarselli et al., 2009; Socher et al., 2011; Li et al., 2015; Tai et al., 2015). Among RNN based methods, many are only applicable to special structured graph, e.g., sequences or trees, (Socher et al., 2011; Tai et al., 2015). One class of models which are applicable to general graphs are so-called graph neural networks (GNNs), Scarselli et al. (2009). The inference procedure is a forward pass that exploits a fixed-length propagation process which resembles synchronous message passing system in the theory of distributed computing, Attiya & Welch (2004). Nodes in the graph have state vectors which are recurrently updated based on their history and received messages. One of the representative work of GNNs, i.e., gated graph neural networks (GGNNs) by Li et al. (2015), uses gated recurrent unit to update the state vectors. Learning such a model can be achieved by the back-propagation through time (BPTT) algorithm or recurrent back-propagation, Chauvin & Rumelhart (1995). It has been shown that GNNs, (Li et al., 2015; 2017; Qi et al., 2017; Gilmer et al., 2017) have a high capacity and achieve state-of-the-art performance in many applications which involve graph-structured data. In this paper, we model the structure of the reinforcement learning agents using GNNs.

**Transfer and Multi-task Learning in RL**   Recently, there has been increased interest in transfer learning tasks for RL, Taylor & Stone (2009), which mainly focus on transferring the policy learned from one environment to another. In Rajeswaran et al. (2017b;a), the authors show that agents in reinforcement learning are prone to over-fitting, and that the learned policies generalize poorly across environments. In model-based RL, traditional control has been well studied for generalization properties, Coros et al. (2010). Gupta et al. (2017) try to increase the transferability via learning invariant visual features. Efforts have also been made from the meta-learning perspective (Duan et al., 2016; Finn et al., 2017b;a). In Wulfmeier et al. (2017), the authors propose a method of transfer learning by using imitation learning. Transferability comes naturally in our model by exploiting the (shared) graph structure of the agents.

Multi-task learning has also received a lot of attention, Wilson et al. (2007). In Teh et al. (2017), the authors use a distilled policy that captures common behaviour across tasks. Wilson et al. (2007); Oh et al. (2017); Andreas et al. (2016) use a hierarchical approach, where multiple sub-policies are learnt. In Yang et al. (2017); Parisotto et al. (2015), the authors exploited shared visual features

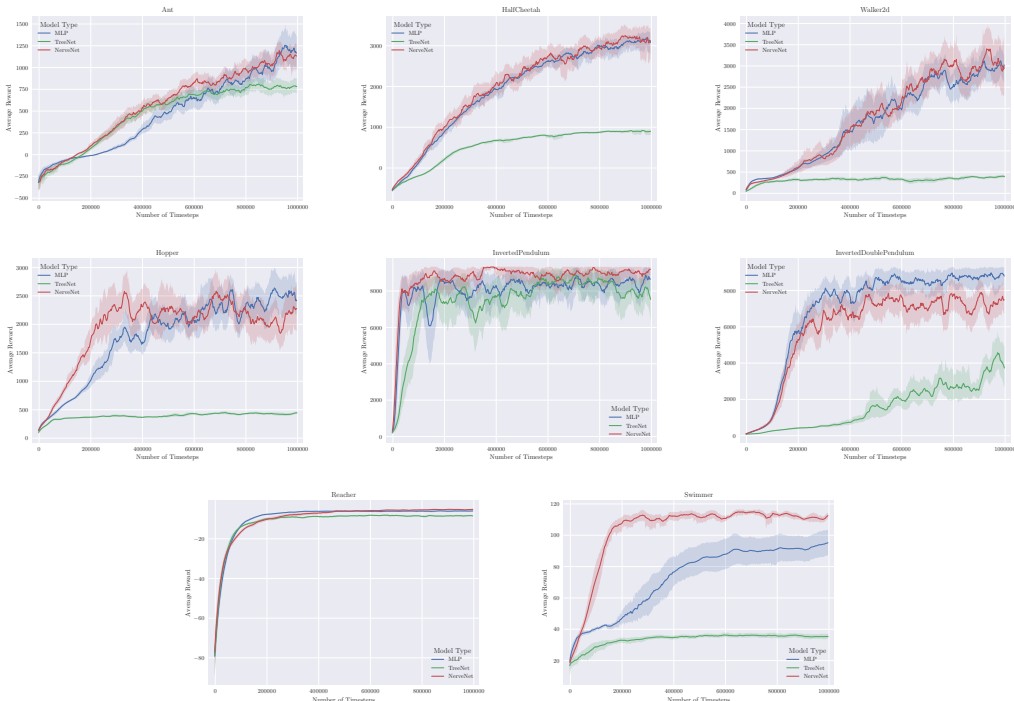

Figure 3: Results of MLP, TreeNet and NerveNet on 8 continuous control benchmarks from the Gym.

to tackle multi-task learning. In Ammar et al. (2014), a multi-task policy gradient is proposed, while, Calandriello et al. (2014) propose multi-task extensions of the fitted Q-iteration algorithm. Successor features in Barreto et al. (2017) are used to boost the performance of multiple tasks. Unlike ours, these methods do not exploit the physical graph structures of the agents.

## 4 EXPERIMENTS

In this section, we first verify the effectiveness of NerveNet on standard MuJoCo environments in OpenAI Gym. We then investigate the transfer abilities of NerveNet and other competitors by customizing some of those environments, as well as the multi-task learning ability and robustness.

### 4.1 COMPARISON ON STANDARD BENCHMARKS OF MUJOCO

**Baselines** We compare NerveNet with the standard MLP models utilized by Schulman et al. (2017) and another baseline which is constructed as follows. We first remove the physical graph structure and introduce an additional super node which connects to all nodes in the graph. This results in a singly rooted depth-1 tree. We refer to this baseline as TreeNet. The propagation model of TreeNet is similar to NerveNet where, however, the policy first aggregates the information from all children and then feeds the state vector of the root to the output model. This simpler model serves as a baseline to verify the importance of the graph structure.

We run experiments on 8 simulated continuous control benchmarks from the Gym, Brockman et al. (2016), which is based on MuJoCo, Todorov et al. (2012). In particular, we use `Reacher`, `InvertedPendulum`, `InvertedDoublePendulum`, `Swimmer`, and four walking or running tasks: `HalfCheetah`, `Hopper`, `Walker2d`, `Ant`. We set the maximum number of training steps to be 1 million for all environments as it is enough to solve them. Note that for `InvertedPendulum`, different from the original one in Gym, we add the distance penalty of the cart and velocity penalty so that the reward is more consistent to the `InvertedDoublePendulum`. This change of design also makes the task more challenging.

Table 1: Performance of the Pre-trained models on `CentipedeFour` and `CentipedeSix`.

| C_Four | Reward Avg | Std | Max | C_Six | Reward Avg | Std | Max |
|--------|-----------|-----|-----|-------|-----------|-----|-----|
| NerveNet | 2799.9 | 1247.2 | 3833.9 | NerveNet | 2507.1 | 738.4 | 2979.2 |
| MLP | 2398.5 | 1390.4 | 3936.3 | MLP | 2793.0 | 1005.2 | 3529.5 |
| TreeNet | 1429.6 | 957.7 | 3021.7 | TreeNet | 2229.7 | 1109.4 | 3727.4 |

**Results** We do grid search to find the best hyperparameters and leave the details in the Appendix 6.3. As the randomness might have a big impact on the performance, for each environment, we run 3 experiments with different random seeds and plot the average curves and the standard deviations. We show the results in Figure 3. From the figures, we can see that MLP with the same setup as in Schulman et al. (2017) works the best in most of tasks.[1] NerveNet basically matches the performance of MLP in terms of sample efficiency as well as the performance after it converges. In most cases, the TreeNet is worse than NerveNet which highlights the importance of keeping the physical graph structure.

## 4.2 STRUCTURE TRANSFER LEARNING

We now benchmark our model in the task of structure transfer learning by creating customized environments based on the existing ones from MuJoCo. We mainly investigate two types of structure transfer learning tasks. The first one is to train a model with an agent of small size (small graph) and apply the learned model to an agent with a larger size, i.e., *size transfer*. When increasing the size of the agent, observation and action space also increase which makes learning more challenging. Another type of structure transfer learning is *disability transfer* where we first learn a model for the original agent and then apply it to the same agent with some components disabled. If one model overfits the environment, disabling some components of the agent might bring catastrophic performance degradation. Note that for both transfer tasks, all factors of environments do not change except the structure of the agent.

**Centipede** We create the first environment in which the agent has a similar structure to a centipede. The goal of the agent is to run as fast as possible along the $y$-direction in the MuJoCo environment. The agent consists of repetitive torso bodies where each one has two legs attached. For two consecutive bodies, we add two actuators which control the rotation between them. Furthermore, each leg consists of a `thigh` and `shin`, which are controlled by two hinge actuators. By linking copies of torso bodies and corresponding legs, we create agents with different lengths. Specifically, the shortest Centipede is `CentipedeFour` and the longest one is `CentipedeFourty` due to the limit of supported resource of MuJoCo. For each time step, the total reward is the speed reward minus the energy cost and force feedback from the ground. Note that in practice, we found that training a `CentipedeEight` from scratch is already very difficult. For *size transfer* experiments, we create many instances which are listed in Figure 4, like "4to06", "6to10". For *disability transfer*, we create `CrippleCentipede` agents of which two back legs are disabled. In Figure 4, `CrippleCentipede` is specified as "Cp".

**Snakes** We also create a snake-like agent which is common in robotics, Crespi & Ijspeert (2008). We design the `Snake` environment based on the `Swimmer` model in Gym. The goal of the agent is to move as fast as possible. For details of the environment, please see the schematic figure 16.

### 4.2.1 EXPERIMENTAL SETTINGS

To fully investigate the performance of NerveNet, we build several baseline models for structure transfer learning which are explained below.

**NerveNet** For the NerveNet, since all the weights are exactly the same for the small and the large-agent models, we directly use the old weights trained on the small-agent model. When the large

---

[1] By applying the adaptive learning rate schedule from Schulman et al. (2017), we obtained better performances than the ones reported in original paper.

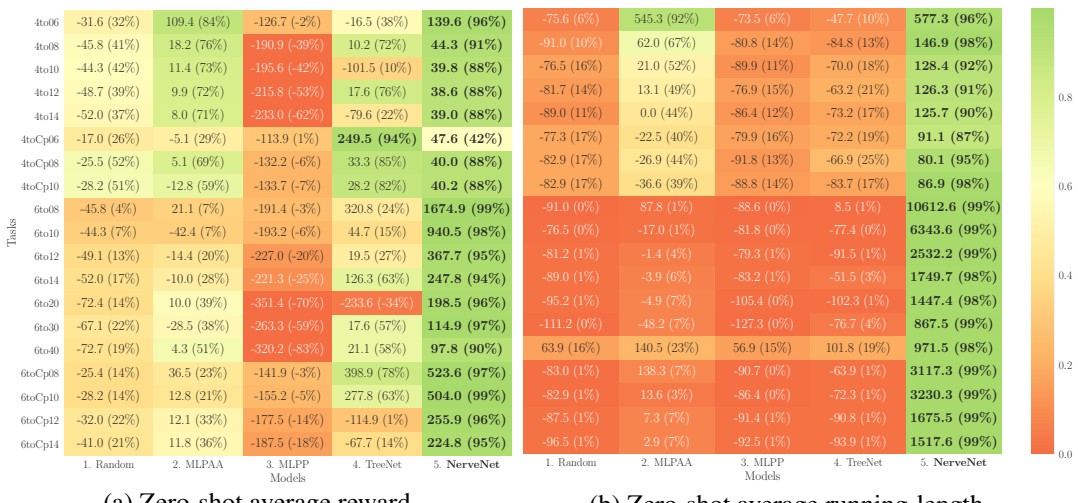

(a) Zero-shot average reward.  (b) Zero-shot average running-length.

Figure 4: Performance of zero-shot learning on centipedes. For each task, we run the policy for 100 episodes and record the average reward and average length the agent runs before falling down.

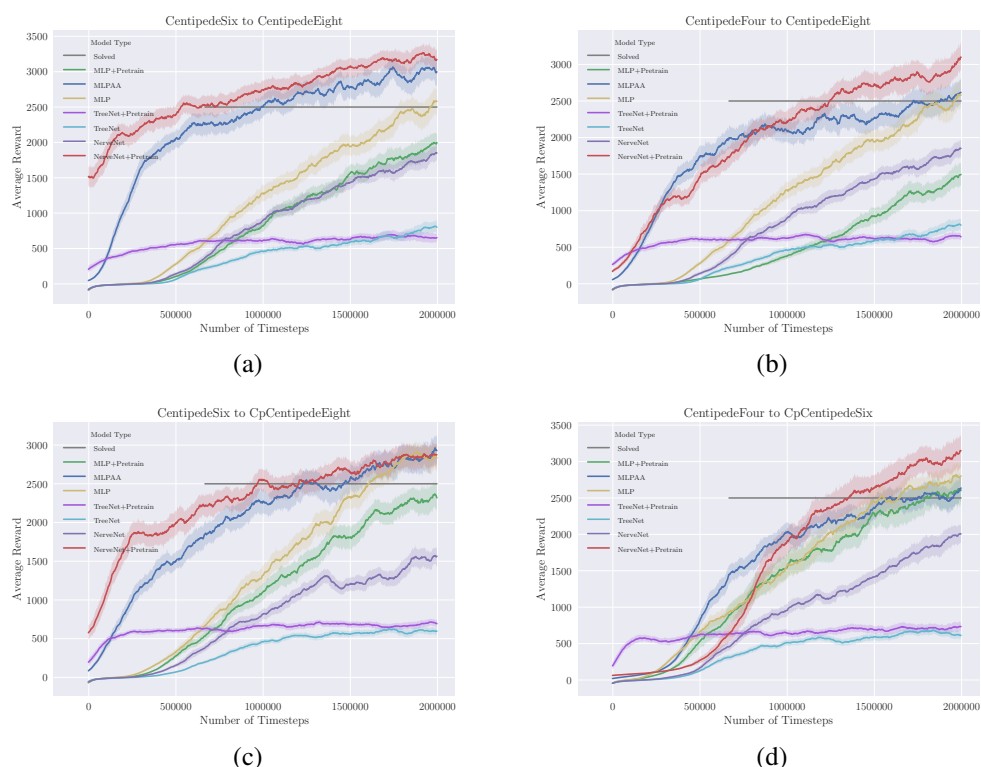

Figure 5: (a), (b): Results of fine-tuning for *size transfer* experiments. (c), (d) Results of fine-tuning for *disability transfer* experiments.

agent has repetitive structure, we further re-use the weights of the corresponding joints from the small-agent model.

**MLP Pre-trained (MLPP)**  For the MLP based model, while transferring from one structure to another, the size of the input layer changes since the size of the observation changes. One straight-

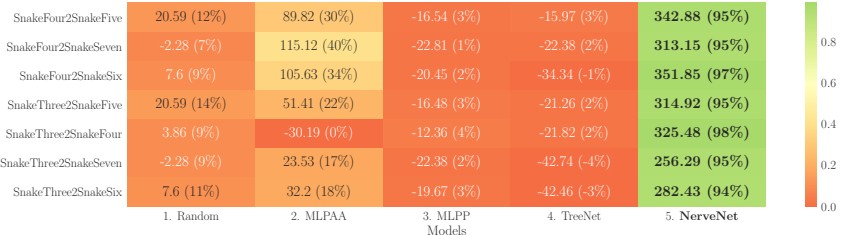

| | 1. Random | 2. MLPAA | 3. MLPP | 4. TreeNet | 5. **NerveNet** |
|---|---|---|---|---|---|
| SnakeFour2SnakeFive | 20.59 (12%) | 89.82 (30%) | -16.54 (3%) | -15.97 (3%) | **342.88 (95%)** |
| SnakeFour2SnakeSeven | -2.28 (7%) | 115.12 (40%) | -22.81 (1%) | -22.38 (2%) | **313.15 (95%)** |
| SnakeFour2SnakeSix | 7.6 (9%) | 105.63 (34%) | -20.45 (2%) | -34.34 (-1%) | **351.85 (97%)** |
| SnakeThree2SnakeFive | 20.59 (14%) | 51.41 (22%) | -16.48 (3%) | -21.26 (2%) | **314.92 (95%)** |
| SnakeThree2SnakeFour | 3.86 (9%) | -30.19 (0%) | -12.36 (4%) | -21.82 (2%) | **325.48 (98%)** |
| SnakeThree2SnakeSeven | -2.28 (9%) | 23.53 (17%) | -22.38 (2%) | -42.74 (-4%) | **256.29 (95%)** |
| SnakeThree2SnakeSix | 7.6 (11%) | 32.2 (18%) | -19.67 (3%) | -42.46 (-3%) | **282.43 (94%)** |

Figure 6: Results on zero-shot transfer learning on snake agents. Each tasks are simulated for 100 episodes.

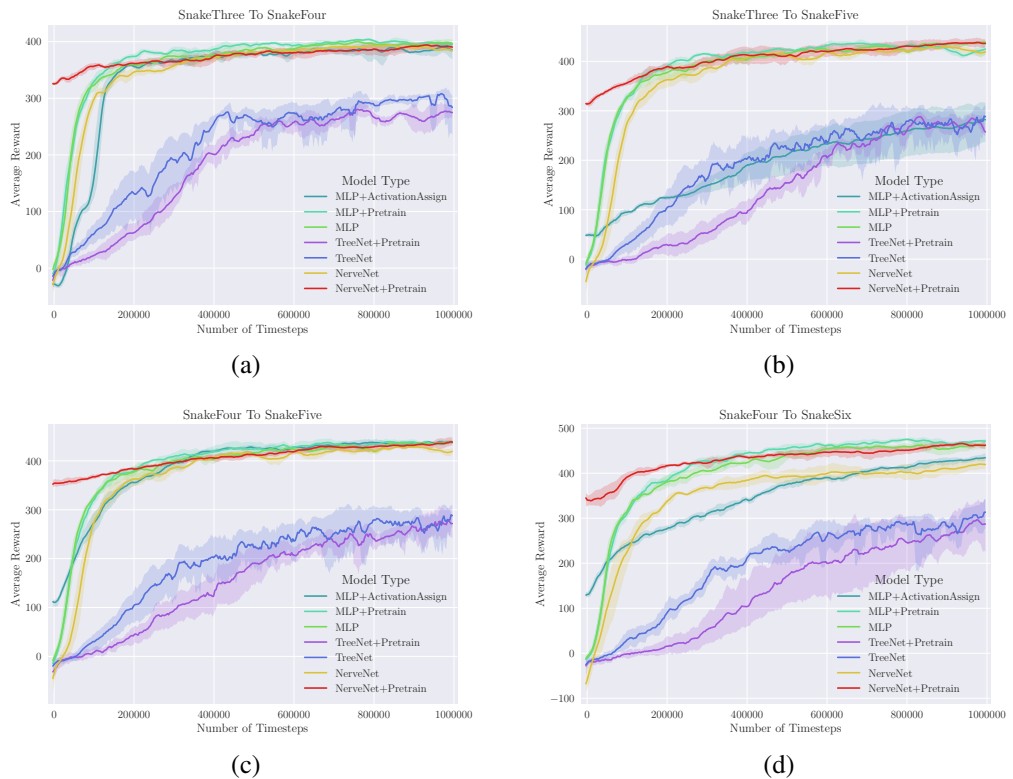

Figure 7: Results of finetuning on snake environments.

forward idea is to reuse the weights from the first hidden layer to the output layer and randomly initialize the weights of the new input layer.

**MLP Activation Assigning (MLPAA)**    Another way of making MLP transferable is assigning the weights of the small-agent model to the corresponding partial weights of the large-agent model and setting the remaining weights to be zero. Note that we do not add or remove any layers from the small-agent model to the large-agent except for changing the size of the layers. By doing so, we can keep the output of the large-agent model to be same as the small-agent in the beginning, i.e., keeping the same initial policy.

**TreeNet**    TreeNet is similar as the model described before. We apply the same way of assigning weights as MLPAA to TreeNet for the transfer learning task.

**Random**    We also include the random policy which is uniformly sampled from the action space.

### 4.2.2 RESULTS

**Centipedes**  For the Centipedes environment, we first run experiments of all models on `CentipedeSix` and `CentipedeFour` to get the pre-trained models for transfer learning. We train different models until these agents run equally well as possible, which is reported in Table 1. Note that, in practice, we train TreeNet on `CentipedeFour` for more than 8 million time steps. However, due to the difficulty of optimizing TreeNet on `CentipedeFour`, the performance is still lower. But visually, the TreeNet agent is able to run in `CentipedeFour`.

We then examine the zero-shot performance where zero-shot means directly applying the model trained with one setting to the other without any fine-tuning. To better visualize the results, we linearly normalize the performance to get a performance score, and color the results accordingly. The normalization scheme is recorded in Appendix 11. The performance score is less than 1, and is shown in the parentheses behind the original results. As we can see from Figure 4 (full chart in Appendix 6.5), NerveNet outperforms all competitors on all settings, except in the 4toCp06 scenario. Note that transferring from `CentipedeFour` is more difficult than from `CentipedeSix` since the situation where one torso connects to two neighboring torsos only happens beyond 4 bodies.

TreeNet has a surprisingly good performance on tasks from `CentipedeFour`. However, by checking the videos, the learned agent is actually not able to "move" as good as other methods. The high reward is mainly due to the fact that TreeNet policy is better at standing still and gaining alive bonus. We argue that the average running-length in each episode is also a very important metric.

By including the results of running-length, we notice that NerveNet is the only model able to walk in the zero-shot setting. In fact, the performance of NerveNet is orders-of-magnitude better, and most of the time, agents from other methods cannot even move forward. We also notice that if transferred from `CentipedeSix`, NerveNet is able to provide walkable pre-trained models on all new agents.

We fine-tune for both *size transfer* and *disability transfer* experiments and show the training curves in Figure 5. From the figure, we can see that by using the pre-trained model, NerveNet significantly decreases the number of episodes required to reach the level of reward which is considered as solved. By looking at the videos, we notice that the bottleneck of learning for the agent is "how to stand". When training from scratch, it can be seen that almost $0.5$ million time steps are spent on a very flat reward surface. Therefore, the MLPAA agents, which copy the learned policy, are able to stand and bypass this time-consuming process and reach to a good performance in the end.

Moreover, by examining the result videos, we noticed that the "walk-cycle" behavior is observed for NerveNet but is not common for others. Walk-cycle are adopted for many insects in the world, Biewener (2003). For example, six-legged ants use a tripedal gait, where the legs are used in two separate triangles alternatively touching the ground. We give more details of walk-cycle in Section 4.5.

One possible reason is that the agent of MLP based method (MLPAA, MLPP) learns a policy that does not utilize all legs. From `CentipedeEight` and up, we do not observe any MLP agents to be able to coordinate all legs whereas almost all policies learned by NerveNet use all legs. Therefore, NerveNet is better at utilizing structure information and not over-fitting the environments.

**Snakes**  The zero-shot performance for snakes is summarized in Figure 6. As we can see, NerveNet has the best performance on all transfer learning tasks. In most cases, NerveNet has a starting reward value of more than 300, which is a pretty good policy since 350 is considered as solved for `snakeThree`. By looking at the videos, we found that agents of other competitors are not able to control the new actuators in the zero-shot setting. They either overfit to the original models, where the policy is completely useless in the new setting (e.g., the MLPAA is worse than random policy in `SnakeThree2SnakeFour`), or the new actuators are not able to coordinate with the old actuators trained before. While for NerveNet, the actuators are able to coordinate to its neighbors, regardless of whether they are new to the agents.

We also summarize the training curves of fine-tuning in Fig. 7. We can observe that NerveNet has a very good initialization with the pre-trained model, and the performance increases with fine-tuning. When training from scratch, NerveNet is less sample efficient compared to the MLP model which might be caused by the fact that optimizing our model is more challenging than MLP. Fine-tuning helps to improve the sample efficiency of our model by a large margin. At the same time, although

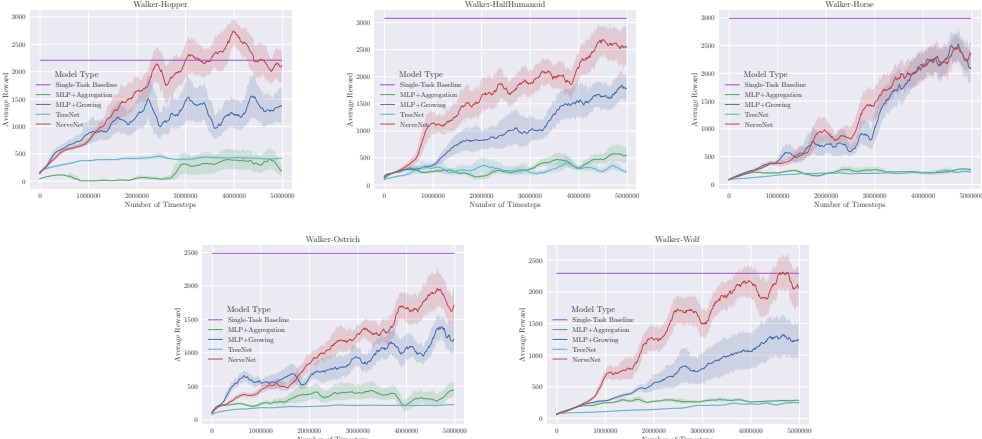

Figure 8: Results of Multi-task learning. We train the networks simultaneously on five different tasks from `Walker`.

| Model | | HalfHumanoid | Hopper | Ostrich | Wolf | Horse | Average |
|---|---|---|---|---|---|---|---|
| MLP | Reward | 1775.75 | 1369.59 | 1198.88 | 1249.23 | 2084.07 | / |
| | Ratio | 57.7% | 62.0% | 48.2% | 54.5% | 69.7% | 58.6% |
| TreeNet | Reward | 237.81 | 417.27 | 224.07 | 247.03 | 223.34 | / |
| | Ratio | 79.3% | 98.0% | 57.4% | 141.2% | 99.2% | 94.8% |
| NerveNet | Reward | 2536.52 | 2113.56 | 1714.63 | 2054.54 | 2343.62 | / |
| | Ratio | 96.3% | 101.8% | 98.8% | 105.9% | 106.4% | 101.8% |

Table 2: Results of Multi-task learning, with comparison to the single-task baselines. For three models, the first row is the mean reward of each model of the last 40 iterations. The second row indicates the percentage of the performance of the multi-task model compared with the single-task baseline of each model.

the MLPAA has a very good initialization, its performance progresses slowly with the increasing number of episodes. In most experiments, the MLPAA and TreeNet did not match the performance of its non-pretrained MLP baseline.

## 4.3 MULTI-TASK LEARNING

In this section, we show that NerveNet has a good potential of multi-task learning by incorporating structure prior into the network structure. It is important to point out that multi-task learning represents a very difficult, and more often the case, unsolved problem in RL. Most multi-task learning algorithms, Teh et al. (2017); Andreas et al. (2016); Oh et al. (2017); Yang et al. (2017) have not been applied to domains as difficult as locomotion for complex physical models, not to mention multi-task learning among different agents with different dynamics.

In this work, we constrain our problem domain, and design the `Walker` multi-task learning task-set, which contains five 2d-walkers. We aim to test the model's ability of multi-task learning, in particular, the ability to control multiple agents using one unified network. The walkers are very different in terms of their dynamics, since they have very distinct structures, different types and numbers of controllers. `Walker-HalfHumanoid` and `Walker-Hopper` are variants of `Walker2d` and `Hopper` from the original MuJoCo Benchmarks, respectively. `Walker-Horse`, `Walker-Ostrich`, `Walker-Wolf` on the other hand, are agents mimicking the natural animals. Just like the real animals, some of the agents have tails or necks to help them to balance. The detailed schematic figures are shown in the Appendix 6.8.

| Model | | Halfhumanoid | Hopper | Wolf | Ostrich | Horse | Average |
|---|---|---|---|---|---|---|---|
| Mass | MLP | 33.28% | 74.04% | 94.68% | 59.23% | 40.61% | 60.37% |
| | NerveNet | 95.87% | 93.24% | 90.13% | 80.2% | 69.23% | 85.73% |
| Strength | MLP | 25.96% | 21.77% | 27.32% | 30.08% | 19.80% | 24.99% |
| | NerveNet | 31.11% | 42.20% | 42.84% | 31.41% | 36.54% | 36.82% |

Table 3: Results of robustness evaluations. Note that we show the average results for each type of parameters after perturbation. And the results are columned by the agent type. The ratio of the average performance of perturbed agents and the original performance is shown in the figure. Details are listed in 6.6.

### 4.3.1 EXPERIMENTAL SETTINGS

To show the ability of multi-task learning of NerveNet, we design several baselines. We use a vanilla multi-task policy update for all models. More specifically, for each sub-task in the multi-task learning task-set, we use an equal number of time steps for each policy's update and calculate the gradients separately. Gradients are then aggregated and the mean value of gradients is applied to update the network. To compensate for the additional difficulty in training more agents and tasks, we linearly increase the number of update epochs during each update in training, as well as the total number of time steps generated before the training is terminated. The hyper-parameter setting is summarized in Appendix 6.7.

**NerveNet** For NerveNet, the weights are naturally shared among different agents. More specifically, for different agents, the weight matrices for propagation and output are shared.

**MLP Sharing** For the MLP method, we shared the weight matrices between hidden layers.

**MLP Aggregation** In the MLP Sharing approach, the total size of the weight matrices grows with the number of tasks. For different agents, whose dimension of the observations are usually different, weights from observation to the first hidden layer cannot be reused in the MLP Sharing approach. Therefore in the MLP Aggregation method, we multiply each element of the observation vector separately by one matrix, and aggregate the resulting vectors from each element. The size of this multiplying matrix is (1, dimension of the first hidden layer).

**TreeNet** Similarly, TreeNet also has the benefits that its weights are naturally shared among different agents. However, TreeNet has no knowledge of the agents' physical structure, where the information of each node is aggregated into the root node.

We also include the baseline of training single-task MLP for each agent. We train the single-task MLP baselines for 1 million time steps per agent. In the Figure 8, we align the results of single-task MLP baseline and the results of multi-task models by the number of episodes of one task.

As can be seen from Figure 8, NerveNet achieves the best performance in all the sub-tasks. In `Walker-HalfHumanoid`, `Walker-Hopper`, `Walker-Ostrich`, `Walker-Wolf` our NerveNet is able to out-perform other agents by a large margin. In `Walker-Horse`, the performance of NerveNet and MLP Sharing are relatively similar. For MLP Sharing, the performance on other four agents are relatively limited, while for `Walker-Hopper`, the improvement of performance is limited from half of the experiment onwards. The MLP Aggregation and TreeNet methods are not able to solve the multi-task learning problem, with both of them stuck at a very low reward level. In the vanilla optimization setting, we show that NerveNet has a bigger potential than the baselines.

From Table 2, one can observe that the performance of MLP drops drastically (42% performance drop) when switching from single-task to multi-task learning, while for NerveNet, there is no obvious drop in performance. Our intuition is that NerveNet is better at learning generalized features, and learning of different agents can help in training other agents, while for MLP methods, the performance decreases due the competition of different agents.

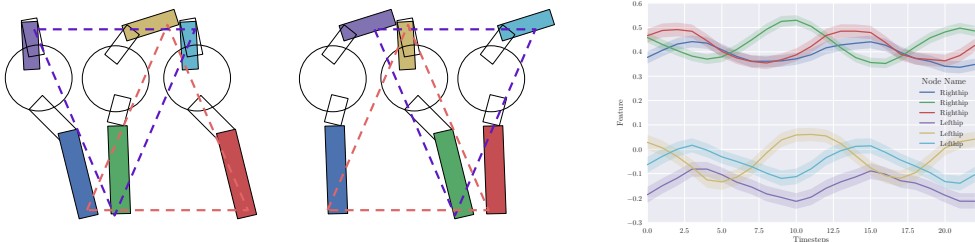

Figure 9: Diagram of the walk cycle. In the left figure, legs within the same triangle are used simultaneously. For each leg, we use the same color for their diagram on the left and their curves on the right.

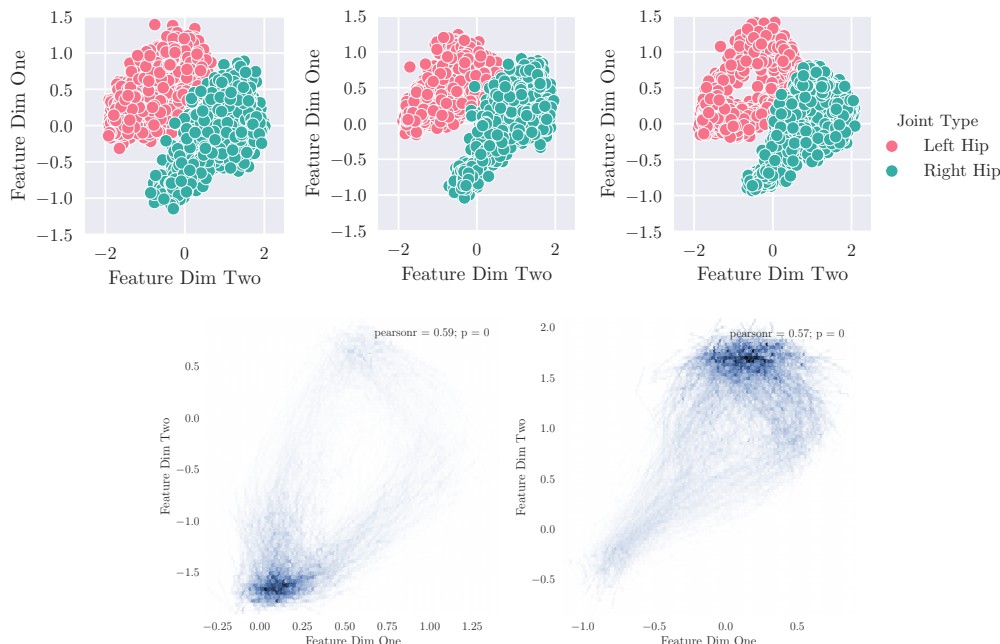

Figure 10: Results of visualization of feature distribution and trajectory density. As can be seen from the figure, NerveNet agent is able to learn shareable features for its legs, and certain walk-cycle is learnt during training.

## 4.4 Robustness of Learnt Policies

In this section, we also report the robustness of our policy by perturbing the agent parameters. In reality, the parameters simulated might be different from the actual parameters of the agents. Therefore, it is important that the agent is robust to parameters perturbation. The model that has the better ability to learn generalized features are likely more robust.

We perturb the mass of the geometries (rigid bodies) in MuJoCo as well as the scale of the forces of the joints. We use the pre-trained models with similar performance on the original task for both the MLP and NerveNet. The performance is tested in five agents from `Walker` task set. The average performance is recorded in Table 3, and the specific details are summarized in Appendix 6.6. The robustness of NerveNet' policy is likely due to the structure prior of the agent instilled in the network, which facilitates overfitting.

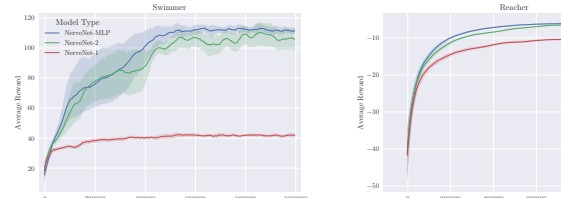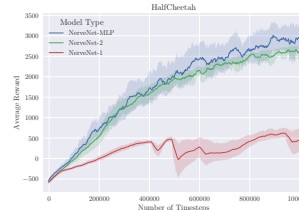

Figure 11: Results of several variants of NerveNet for the reinforcement learning agents.

### 4.5 INTERPRETING THE LEARNED REPRESENTATIONS

In this section, we try to visualize and interpret the learned representations. We extract the final state vectors of nodes of NerveNet trained on `CentipedeEight`. We then apply 1-$D$ and 2-$D$ PCA on the node representations. In Figure 10, we notice that each pair of legs is able to learn invariant representations, despite their different position in the agent. We further plot the trajectory density map in the feature map. By recording the period of the walk-cycle, we plot the transformed features of the 6 legs on Figure 9. As we can see, there is a clear periodic behavior of our hidden representations learned by our model. Furthermore, the representations of adjacent left legs and the adjacent right legs demonstrate a phase shift, which further proves that our agents are able to learn the walk-cycle without any additional supervision.

### 4.6 COMPARISON OF MODEL VARIANTS

We have several variants of NerveNet, based on the type of network we use for the policy/value representation. We here compare all variants. Again, we run experiments for each task three times. The details of hyper-parameters are given in the Appendix. For each environment, we train the network for one million time steps, with batch size 2050 for one update.

As we can see from Figure 11, the NerveNet-MLP and NerveNet-2 variants perform better than NerveNet-1. One potential reason is that sharing the weights of the value and policy networks makes the trust-region based optimization methods, like PPO, more sensitive to the weight $\alpha$ of the value function in equation 8. Based on the figure, choosing $\alpha$ to be 1 is not giving good performance on the tasks we experimented on.

## 5 CONCLUSION

In this paper, we aimed to exploit the body structure of Reinforcement Learning agents in the form of graphs. We introduced a novel model called NerveNet which uses a Graph Neural Network to represent the agent's policy. At each time instance of the environment, NerveNet takes observations for each of the body joints, and propagates information between them using non-linear messages computed with a neural network. Propagation is done through the edges which represent natural dependencies between joints, such as physical connectivity. We experimentally showed that our NerveNet achieves comparable performance to state-of-the-art methods on standard MuJoCo environments. We further propose our customized reinforcement learning environments for benchmarking two types of structure transfer learning tasks, i.e., *size* and *disability transfer*. We demonstrate that policies learned by NerveNet are significantly better than policies learned by other models and are able to transfer even in a zero-shot setting.

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

# 6 APPENDIX

## 6.1 DETAILS OF NERVENET

We use MLP to compute the messages which uses $tanh$ nonlinearities as the activation function. We do a grid search on the size of the MLP to compute the messages, the details of which are listed in Table 5, 4.

Throughout all of our experiments, we use average aggregation and GRU as the update function.

Table 4: Parameters used during training.

| Parameters | Value Set | Parameters | Value Set |
|---|---|---|---|
| Value Discount Factor $\gamma$ | .99 | GAE $\lambda$ | .95 |
| PPO Clip Value | 0.2 | Starting Learning Rate | 3e-4 |
| Gradient Clip Value | 5.0 | Target KL | 0.01 |

## 6.2 GRAPH OF AGENT

In MuJoCo, we observe that most `body` nodes are paired with one and only one `joint` node. Thus, we simply merge the two paired nodes into one. We point out that this model is very compact, and is the standard graph we use in our experiments.

In the Gym environments, observation for the `joint` nodes normally includes the angular velocity, twist angle and optionally the torque for the hinge `joint`, and position information for the positional `joint`. For the `body` nodes, velocity, inertia, and force are common observations. For example in the centipede environment 1, the `LeftHip` node will receive the angular velocity $\varpi_j$, and the twist angle $\theta_j$.

## 6.3 HYPERPARAMETER SEARCH

For MLP, we run grid search with the hidden size from two layers to three layers, and with hidden size from 32 to 256. For NerveNet, to reduce the time spent on grid search, we constrain the propagation network and output network to be the same shape. Similarly, we run grid search with the network's hidden size, and at the same time, we run a grid search on the size of node's hidden states from 32 to 64. For the TreeNet, we run similar grid search on the node's hidden states and output network's shape.

For details of hyperparameter search, please see Table 5, 7, 6.

Table 5: Hyperparameter grid search options for MLP.

| MLP | Value Tried |
|---|---|
| Network Shape | [64, 64], [128,128], [256, 256], [64,64,64] |
| Number of Iteration Per Update | 10, 20 |
| Use KL Penalty | Yes, No |
| Learning Rate Scheduler | Linear Decay, Adaptive, Constant |

Table 6: Hyperparameter grid search options for TreeNet.

| TreeNet | Value Tried |
|---|---|
| Network Shape | [64, 64], [128,128], [256, 256] |
| Number of Iteration Per Update | 10, 20 |
| Use KL Penalty | Yes, No |
| Learning Rate Scheduler | Linear Decay, Adaptive, Constant |

Table 7: Hyperparameter grid search options for NerveNet.

| NerveNet | Value Tried |
| --- | --- |
| Network Shape | [64, 64], [128,128], [256, 256] |
| Number of Iteration Per Update | 10, 20 |
| Use KL Penalty | Yes, No |
| Learning Rate Scheduler | Linear Decay, Adaptive, Constant |
| Number of Propogation Steps | 3, 4, 5, 6 |
| NerveNet Variants | NerveNet-1, NerveNet-2, NerveNet-MLP |
| Size of Nodes' Hidden State | 32, 64, 128 |
| Merge `joint` and `body` Node | Yes, No |
| Output Network | Shared, Separate |
| Disable Edge Type | Yes, No |
| Add Skip-connection from / to `root` | Yes, No |

## 6.4 SCHEMATIC FIGURES OF THE PARSED AGENTS

In this section, we also plot the schematic figures of the agents for readers' reference. Graph structures are automatically parsed from the MuJoCo XML configuration files.

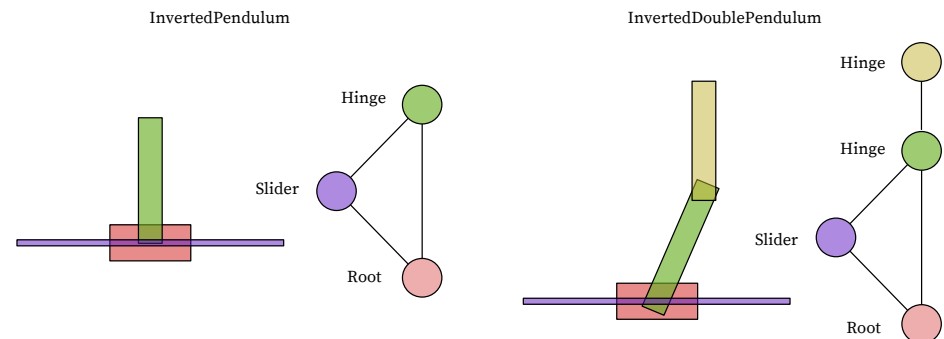

Figure 12: Schematic diagrams and auto-parsed graph structures of the `InvertedPendulum` and `InvertedDoublePendulum`.

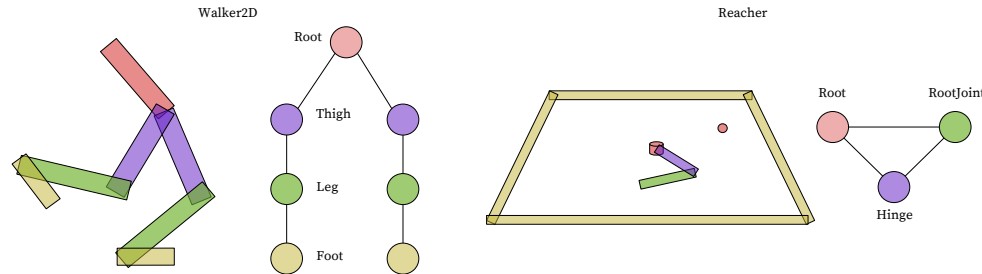

Figure 13: Schematic diagrams and auto-parsed graph structures of the `Walker2d` and `Reacher`.

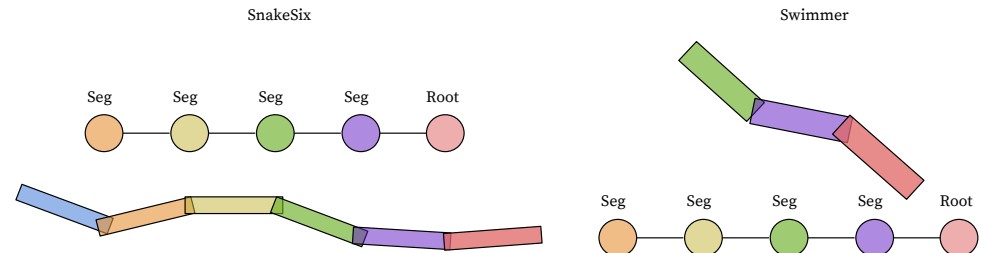

Figure 14: Schematic diagrams and auto-parsed graph structures of the `SnakeSix` and `Swimmer`.

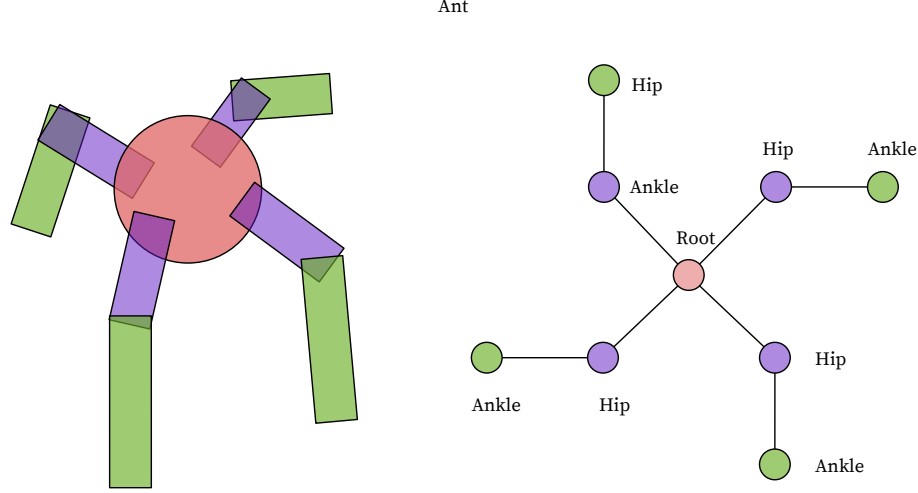

Figure 15: Schematic diagrams and auto-parsed graph structures of the `Ant`.

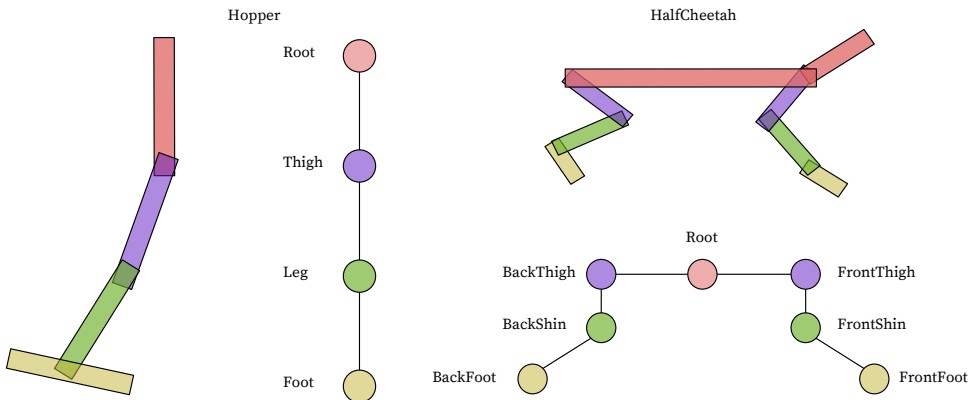

Figure 16: Schematic diagrams and auto-parsed graph structures of the `Hopper` and `HalfCheetah`.

## 6.5 DETAILS OF ZERO-SHOT LEARNING RESULTS

Table 8: Results of the zero-shot learning of Centipede

| Task | Reward Max | | | | | Reward Std | | | | | Reward Average | | | | |
|---|---|---|---|---|---|---|---|---|---|---|---|---|---|---|---|
| | Random | MLPAA | MLPP | TreeNet | NerveNet | Random | MLPAA | MLPP | TreeNet | NerveNet | Random | MLPAA | MLPP | TreeNet | NerveNet |
| CentipedeFour2CentipedeSix | -5.7 | 396.4 | -23.7 | 100.4 | 561.4 | 20.3 | 86.2 | 74 | 73.6 | 125.9 | -31.6 | 109.4 | -126.7 | -16.5 | 139.6 |
| CentipedeFour2CentipedeEight | -10.3 | 107.1 | -33.7 | 84.7 | 161.6 | 30.7 | 19.5 | 146.3 | 17.8 | 35.2 | -45.8 | 18.2 | -190.9 | 10.2 | 44.3 |
| CentipedeFour2CentipedeTen | -5.5 | 55.7 | -47 | 510.1 | 136.7 | 30.3 | 13.3 | 132 | 228.5 | 28.7 | -44.3 | 11.4 | -195.6 | -101.5 | 39.8 |
| CentipedeFour2CentipedeTwelve | -9.9 | 45.2 | -24.8 | 84.8 | 147.7 | 30.5 | 11.9 | 173.2 | 14.9 | 27.4 | -48.7 | 9.9 | -215.8 | 17.6 | 38.6 |
| CentipedeFour2CentipedeFourteen | -10.5 | 46.9 | -21.8 | 550 | 142.8 | 38.1 | 11.8 | 170.9 | 217.6 | 31.2 | -52 | 8 | -233 | -79.6 | 39 |
| CentipedeFour2CentipedeTwenty | -11.9 | 59.6 | -31.8 | 420 | 176.8 | 52.2 | 13.4 | 299.8 | 84.7 | 37.2 | -72.4 | 5.8 | -353.4 | 158.4 | 39 |
| CentipedeFour2CentipedeThirty | 1.4 | 44.3 | -5.8 | 169.4 | 137.9 | 64.8 | 10.8 | 262.2 | 29.2 | 35 | -67.6 | -1.1 | -289.8 | 30 | 30.4 |
| CentipedeFour2CentipedeForty | -0.9 | 22 | -3.7 | 105.1 | 244.1 | 64.9 | 8.4 | 220.2 | 190.8 | 32.4 | -71.5 | -4.3 | -272 | -56.2 | 24.3 |
| CentipedeSix2CentipedeEight | -10.3 | 92.9 | -31 | 394.9 | 2586.6 | 30.7 | 16.1 | 127.4 | 75.2 | 884.4 | -45.8 | 21.1 | -191.4 | 320.8 | 1674.9 |
| CentipedeSix2CentipedeTen | -5.5 | 29.4 | -29.6 | 257 | 2154.9 | 30.3 | 115.7 | 157.2 | 76.4 | 695 | -44.3 | -42.4 | -193.2 | 44.7 | 940.5 |
| CentipedeSix2CentipedeTwelve | -9.9 | 29.6 | -40.2 | 157.1 | 1421.3 | 30.5 | 64.6 | 177.8 | 30.9 | 351.6 | -49.1 | -44.4 | -227 | 19.5 | 367.7 |
| CentipedeSix2CentipedeFourteen | -10.5 | 33.3 | -27.6 | 146.4 | 1150.2 | 38.1 | 108.5 | 141.3 | 9.9 | 249 | -52 | -10 | -221.3 | 126.3 | 247.8 |
| CentipedeSix2CentipedeTwenty | -11.9 | 39.6 | -29.1 | 163.5 | 1452 | 52.2 | 10.1 | 379 | 258.9 | 197.6 | -72.4 | 10 | -351.4 | -233.6 | 198.5 |
| CentipedeSix2CentipedeThirty | 1.4 | 29.3 | -4.3 | 80.2 | 614.3 | 64.6 | 158.2 | 225.3 | 16.2 | 124.3 | -67.1 | -28.5 | -263.3 | 17.6 | 114.9 |
| CentipedeSix2CentipedeForty | -0.9 | 27 | -5.1 | 87.3 | 575.4 | 65.1 | 7.9 | 288.6 | 18.6 | 114.2 | -72.7 | 4.3 | -320.2 | 21.1 | 97.8 |
| CentipedeFour2CpCentipedeSix | 7.9 | 44.2 | -30.2 | 729.8 | 175.1 | 12 | 44.5 | 63.5 | 242.8 | 30.4 | -17 | -5.1 | -113.9 | 249.5 | 47.6 |
| CentipedeFour2CpCentipedeEight | 0.1 | 23.3 | -23.7 | 113.3 | 115 | 14 | 4.5 | 64.6 | 22.8 | 24.1 | -25.5 | -12.8 | -132.2 | 33.3 | 40 |
| CentipedeFour2CpCentipedeTen | -6.6 | 27.8 | -29.9 | 141.4 | 96.6 | 16 | 50.6 | 80.8 | 22.8 | 22.3 | -28.2 | 5.1 | -133.7 | 28.2 | 40.2 |
| CentipedeFour2CpCentipedeTwelve | -6.2 | 27.1 | -44 | 665.6 | 84.6 | 14.9 | 5.7 | 87.9 | 106.8 | 18.7 | -32.2 | 4.4 | -174 | 578.8 | 34.8 |
| CentipedeFour2CpCentipedeFourteen | -11.6 | 25.5 | -24.8 | 383 | 101.9 | 20.5 | 5.7 | 99.8 | 82.9 | 21.8 | -41 | 3.8 | -194 | 157.9 | 33.9 |
| CentipedeSix2CpCentipedeEight | 0.1 | 169.1 | -37.1 | 528.9 | 2143.2 | 13.9 | 31.8 | 76.2 | 152.8 | 494.6 | -25.4 | 36.5 | -141.9 | 398.9 | 523.6 |
| CentipedeSix2CpCentipedeTen | -6.6 | 55.8 | -29.9 | 389.7 | 1904.6 | 16 | 8.9 | 81.9 | 112.9 | 444.5 | -28.2 | 12.8 | -155.2 | 277.8 | 504 |
| CentipedeSix2CpCentipedeTwelve | -6.2 | 49.2 | -34.7 | 343.4 | 1429.4 | 15 | 8.6 | 88.3 | 177.8 | 234.7 | -32 | 12.1 | -177.5 | -114.9 | 255.9 |
| CentipedeSix2CpCentipedeFourteen | -11.6 | 48 | -27.7 | 171.4 | 976.2 | 20.5 | 8.3 | 104.5 | 136.2 | 190.2 | -41 | 11.8 | -187.5 | -67.7 | 224.8 |

| Task | Distance Max | | | | | Distance Std | | | | | Distance Average | | | | |
|---|---|---|---|---|---|---|---|---|---|---|---|---|---|---|---|
| | Random | MLPAA | MLPP | TreeNet | NerveNet | Random | MLPAA | MLPP | TreeNet | NerveNet | Random | MLPAA | MLPP | TreeNet | NerveNet |
| CentipedeFour2CentipedeSix | 63.3 | 2082.6 | 40.5 | 61 | 2523.5 | 55.1 | 460.7 | 48.9 | 35.5 | 578.8 | -75.6 | 545.3 | -73.5 | -47.7 | 577.3 |
| CentipedeFour2CentipedeEight | 70.5 | 528.1 | 77.8 | -47.9 | 676.4 | 60.8 | 101 | 68.7 | 13.7 | 164.6 | -91 | 62 | -80.8 | -84.8 | 146.9 |
| CentipedeFour2CentipedeTen | 29.5 | 283.2 | 104.5 | 79.9 | 580.2 | 51.8 | 69 | 56.3 | 54.5 | 136 | -76.5 | 21 | -89.9 | -70 | 128.4 |
| CentipedeFour2CentipedeTwelve | 31.3 | 179.8 | 43.1 | -34.3 | 674.6 | 45.8 | 58.1 | 47.9 | 12.4 | 133.7 | -81.7 | 13.1 | -76.9 | -63.2 | 126.3 |
| CentipedeFour2CentipedeFourteen | 20.1 | 188.8 | 131.4 | 140.1 | 660.5 | 50.4 | 57.5 | 59.6 | 51.4 | 151.8 | -89 | 0 | -86.4 | -73.2 | 125.7 |
| CentipedeFour2CentipedeTwenty | 59.7 | 191.3 | 53.4 | -3.9 | 852.2 | 46.4 | 57 | 53.2 | 16.1 | 188.5 | -95.2 | -16.8 | -98.4 | -44.2 | 131.8 |
| CentipedeFour2CentipedeThirty | -7.5 | 159.9 | -28.5 | -40.7 | 675.4 | 47.5 | 47.2 | 58.8 | 25.2 | 181.3 | -111.9 | -60.5 | -122.5 | -84.4 | 82 |
| CentipedeFour2CentipedeForty | 166.5 | 194.5 | 162.2 | 156.5 | 1332.4 | 61.9 | 37.2 | 59.3 | 42.7 | 160.9 | 64 | 114.9 | 60.3 | 90.8 | 247.5 |
| CentipedeSix2CentipedeEight | 70.5 | 461.9 | 109.7 | 73.8 | 16225.5 | 60.8 | 93.3 | 65.7 | 26.4 | 5606.5 | -91 | 87.8 | -88.6 | 8.5 | 10612.6 |
| CentipedeSix2CentipedeTen | 29.5 | 101.6 | 13.1 | -36.5 | 14438.9 | 51.8 | 58.7 | 43.8 | 15.9 | 4669.9 | -76.5 | -17 | -81.8 | -77.4 | 6343.6 |
| CentipedeSix2CentipedeTwelve | 31.3 | 130 | 163.9 | -53.4 | 10110.1 | 46 | 51.2 | 59.8 | 13.4 | 2409.6 | -81.2 | -1.4 | -79.3 | -91.5 | 2532.2 |
| CentipedeSix2CentipedeFourteen | 20.1 | 121.6 | 25.3 | -5.2 | 8031.4 | 50.4 | 54.7 | 53.7 | 19.4 | 1748.2 | -89 | -3.9 | -83.2 | -51.5 | 1749.7 |
| CentipedeSix2CentipedeTwenty | 59.7 | 130.9 | 3.8 | 70.3 | 10422.7 | 46.4 | 46.1 | 58.3 | 51.3 | 1415.8 | -95.2 | -4.9 | -105.4 | -102.3 | 1447.4 |
| CentipedeSix2CentipedeThirty | -7.5 | 57.9 | 158.7 | -41.1 | 4693.1 | 61.7 | 48.7 | 52.9 | 21.1 | 941.7 | -111.2 | -48.2 | -127.3 | -76.7 | 867.5 |
| CentipedeSix2CentipedeForty | 166.5 | 232 | 60.6 | 156.9 | 4769.4 | 50.8 | 33.7 | 64.3 | 27.3 | 884.5 | 63.9 | 140.5 | 56.9 | 101.8 | 971.5 |
| CentipedeFour2CpCentipedeSix | 61.9 | 122.4 | 35.7 | 36.8 | 675.4 | 41 | 35.1 | 57.4 | 42.3 | 126 | -77.3 | -22.5 | -79.9 | -72.2 | 91.1 |
| CentipedeFour2CpCentipedeEight | 27.7 | 49.1 | 0 | -5.1 | 408.7 | 40.5 | 16.2 | 57.5 | 27.1 | 101.7 | -82.9 | -26.9 | -91.8 | -66.9 | 80.1 |
| CentipedeFour2CpCentipedeTen | 18.3 | 61.1 | 57.8 | -53.9 | 363 | 39.2 | 31.7 | 35.5 | 12 | 95.4 | -82.9 | -36.6 | -88.8 | -83.7 | 86.9 |
| CentipedeFour2CpCentipedeTwelve | 1.9 | 15.3 | 43.6 | 10.8 | 299.6 | 42 | 21 | 49.9 | 51.2 | 76.2 | -87.9 | -30.4 | -98.3 | -62.2 | 72 |
| CentipedeFour2CpCentipedeFourteen | -1.1 | 23.5 | 66.7 | -43.8 | 349.2 | 41.1 | 19.8 | 48.3 | 16 | 93.7 | -96.5 | -33.5 | -92.5 | -76.3 | 74.8 |
| CentipedeSix2CpCentipedeEight | 27.7 | 788.5 | 16.2 | -23.3 | 12628.5 | 40.5 | 157.2 | 58.6 | 18.1 | 2939.4 | -83 | 138.3 | -90.7 | -63.9 | 3117.3 |
| CentipedeSix2CpCentipedeTen | 18.3 | 215 | 97.2 | -28.4 | 12256.1 | 39.1 | 42.7 | 46.6 | 16.9 | 2844.5 | -82.9 | 13.6 | -86.4 | -72.3 | 3230.3 |
| CentipedeSix2CpCentipedeTwelve | 1.9 | 209.6 | 22.8 | 52.7 | 9370.8 | 42 | 41.4 | 46.5 | 46.9 | 1538.9 | -87.5 | 7.3 | -91.4 | -90.8 | 1675.5 |
| CentipedeSix2CpCentipedeFourteen | -1.1 | 148.8 | | 11.3 | 6456.2 | 42 | 38 | 46.5 | 33.6 | 1293.4 | -96.5 | 2.9 | -92.5 | -93.9 | 1517.6 |

Table 9: Results of the zero-shot learning of `Centipede` using MLP-Bind method.

| Task | Distance Max | Distance Std | Distance Average | Reward Max | Reward Std | Reward Average |
|---|---|---|---|---|---|---|
| CentipedeFour2CentipedeSix | 7.09 | 157.46 | 141.51 | 274.99 | 52.26 | 62.13 |
| CentipedeFour2CentipedeEight | 2.8 | 53.23 | -2 | 144.44 | 21.83 | 24.62 |
| CentipedeFour2CentipedeTen | 1.32 | 38.12 | -5.84 | 256.34 | 31.45 | 30.06 |
| CentipedeFour2CentipedeTwelve | 0.83 | 36.14 | -26.92 | 160.02 | 30.8 | 28.79 |
| CentipedeFour2CentipedeFourteen | 1.28 | 39.76 | -17.9 | 323.97 | 46.03 | 41.41 |
| CentipedeFour2CentipedeTwenty | 1.07 | 39.98 | -34.45 | 527.04 | 76.31 | 44.94 |
| CentipedeFour2CentipedeThirty | 0.51 | 34.58 | -56.76 | 565.61 | 82.94 | 36.53 |
| CentipedeFour2CentipedeForty | 2.1 | 38.05 | 127.39 | 484.13 | 57.71 | 27.76 |
| CentipedeSix2CentipedeEight | 53.73 | 1073.33 | 1158.73 | 1070.86 | 209.24 | 235.97 |
| CentipedeSix2CentipedeTen | 1.94 | 50.27 | 33.14 | 63.05 | 11.88 | 18.65 |
| CentipedeSix2CentipedeTwelve | 1.76 | 44.27 | 9.4 | 60.63 | 10.68 | 14.92 |
| CentipedeSix2CentipedeFourteen | 2 | 47.81 | 8.09 | 60.44 | 12.21 | 15.69 |
| CentipedeSix2CentipedeTwenty | 1.07 | 38.46 | -10.27 | 49.71 | 11.42 | 14.25 |
| CentipedeSix2CentipedeThirty | 0.93 | 43.25 | -37.93 | 55.97 | 13.68 | 10.74 |
| CentipedeSix2CentipedeForty | 2.28 | 37.6 | 140.06 | 62.63 | 10.55 | 6.63 |
| CentipedeFour2CpCentipedeSix | 2.03 | 35.71 | -19.01 | 71.5 | 13.57 | 11.47 |
| CentipedeFour2CpCentipedeEight | 0.5 | 23.52 | -25.71 | 85.42 | 10.71 | 7.34 |
| CentipedeFour2CpCentipedeTen | 0.28 | 18.64 | -26.68 | 32.86 | 7.64 | 7.74 |
| CentipedeFour2CpCentipedeTwelve | 0.1 | 15.99 | -29.97 | 50.4 | 7.98 | 6.35 |
| CentipedeFour2CpCentipedeFourteen | 0.23 | 18.74 | -30.56 | 50.78 | 9.74 | 7.52 |
| CentipedeSix2CpCentipedeEight | 4.43 | 90.32 | 39.57 | 162.91 | 29.33 | 29.09 |
| CentipedeSix2CpCentipedeTen | 0.65 | 25.08 | -23.72 | 41.9 | 7.38 | 8.82 |
| CentipedeSix2CpCentipedeTwelve | 0.6 | 26.51 | -27.88 | 30.09 | 7.2 | 8.32 |
| CentipedeSix2CpCentipedeFourteen | 0.67 | 28.85 | -20.84 | 43.44 | 8.66 | 11.02 |

In the MLP-Bind method, we bind the weights of MLP. By doing this, the weights of the agent from the similar structures will be shared. For example, in the `centipede` environment, the weights from observation to action of all the LeftHips are constrained to be same.

Table 10: Results of the zero-shot learning of `Snake`

| | Reward Min | | | | |
| --- | --- | --- | --- | --- | --- |
| | Random | MLPAA | MLPP | TreeNet | NerveNet |
| SnakeFour-v1 | -31.719 | -43.328 | -64.958 | -58.811 | 308.994 |
| SnakeFive-v1 | -56.382 | 29.262 | -55.562 | -39.346 | 301.4 |
| SnakeSix-v1 | -44.888 | -5.306 | -42.744 | -76.42 | 266.066 |
| SnakeSeven-v1 | -52.764 | -25.511 | -49.765 | -59.52 | 231.557 |
| SnakeFive-v1 | -56.382 | -30.466 | -49.953 | -47.339 | 329.456 |
| SnakeSix-v1 | -44.888 | -35.655 | -52.18 | -90.837 | 333.237 |
| SnakeSeven-v1 | -52.764 | 89.358 | -57.334 | -52.015 | 253.828 |
| | Reward Std | | | | |
| | Random | MLPAA | MLPP | TreeNet | NerveNet |
| SnakeFour-v1 | 9.544 | 5.792 | 10.769 | 10.146 | 5.513 |
| SnakeFive-v1 | 13.318 | 6.191 | 11.715 | 7.883 | 5.788 |
| SnakeSix-v1 | 11.515 | 13.535 | 10.304 | 16.869 | 7.395 |
| SnakeSeven-v1 | 11.21 | 12.131 | 10.981 | 6.953 | 7.179 |
| SnakeFive-v1 | 13.318 | 52.71 | 11.544 | 8.519 | 5.85 |
| SnakeSix-v1 | 11.515 | 64.549 | 10.528 | 23.109 | 6.604 |
| SnakeSeven-v1 | 11.21 | 10.683 | 11.615 | 13.529 | 14.084 |
| | Reward Max | | | | |
| | Random | MLPAA | MLPP | TreeNet | NerveNet |
| SnakeFour-v1 | 3.863 | -19.409 | 10.118 | 0.918 | 338.374 |
| SnakeFive-v1 | 20.585 | 63.866 | 3.136 | -2.329 | 326.974 |
| SnakeSix-v1 | 7.6 | 58.779 | 3.51 | -2.139 | 305.011 |
| SnakeSeven-v1 | -2.28 | 51.265 | 1.325 | -28.087 | 275.536 |
| SnakeFive-v1 | 20.585 | 139.617 | 5.305 | 1.487 | 356.366 |
| SnakeSix-v1 | 7.6 | 156.771 | 2.316 | -4.159 | 367.768 |
| SnakeSeven-v1 | -2.28 | 143.824 | 2.596 | 8.394 | 336.561 |
| | Reward Average | | | | |
| | Random | MLPAA | MLPP | TreeNet | NerveNet |
| SnakeFour-v1 | -13.531 | -30.185 | -12.363 | -21.824 | 325.476 |
| SnakeFive-v1 | -17.829 | 51.411 | -16.481 | -21.256 | 314.919 |
| SnakeSix-v1 | -19.89 | 32.2 | -19.674 | -42.461 | 282.426 |
| SnakeSeven-v1 | -22.099 | 23.533 | -22.378 | -42.742 | 256.293 |
| SnakeFive-v1 | -17.829 | 89.823 | -16.54 | -15.975 | 342.881 |
| SnakeSix-v1 | -19.89 | 105.632 | -20.454 | -34.343 | 351.85 |
| SnakeSeven-v1 | -22.099 | 115.123 | -22.81 | -22.383 | 313.149 |

### 6.5.1 LINEAR NORMALIZATION ZERO-SHOT RESULTS AND COLORIZATION.

As the scale of zero-shot results is very different, we normalize the results across different models for each transfer learning task. For each task, we record the worst value of results from different models and the pre-set worst value $V_{\min}$. we set the normalization minimun value as this worst value. We calculate the normalization maximum value by $\lfloor \max(V)/\text{IntLen} \rfloor * \text{IntLen}$.

Table 11: Parameters for linear normalization during zero-shot results. Results are shown in the order of Centipedes' reward, Centipedes' running-length, Snakes' reward

| parameters | $V_{\min}$ | IntLen |
| --- | --- | --- |
| value | -100, -100, -20 | 30, 30, 30 |

## 6.6 DETAILS OF ROBUSTNESS RESULTS

| Model | Mass | | | | |
|---|---|---|---|---|---|
| | Hopper | Halfhumanoid | Horse | Wolf | Ostrich |
| NerveNet | 1981.59 | 2519.6 | 2304.25 | 1862.21 | 922.12 |
| | 2047.92 | 2392.05 | 1900.43 | 1955.16 | 932.48 |
| | 1968.87 | 2282.84 | 1703.47 | 1648.28 | 983.11 |
| | 2062.36 | 2161.61 | 1239.72 | 1576.21 | 910.29 |
| | 1852.76 | 1804.52 | 946.08 | 1354.51 | 791.84 |
| MLP | 2268.64 | 421.8 | 1692.08 | 1932.64 | 878.75 |
| | 1932.98 | 2115.75 | 1083.62 | 1754.94 | 945.94 |
| | 1637.47 | 278.92 | 690.22 | 1459.93 | 879.66 |
| | 1591.51 | 1224.98 | 538.14 | 1039.9 | 805.99 |
| | 1073.1 | 353.56 | 443.92 | 814.55 | 802.41 |
| Model | Strength | | | | |
| | Hopper | Halfhumanoid | Horse | Wolf | Ostrich |
| NerveNet | 1961.62 | 1528.89 | 2316.82 | 1827.73 | 794.56 |
| | 1117.91 | 940.96 | 960.34 | 862.19 | 539.45 |
| | 492.33 | 594.53 | 488.51 | 274.77 | 371.19 |
| | 468.4 | 308.08 | 305.16 | 189.49 | 255.12 |
| | 446.14 | 249.6 | 201.42 | 133.84 | 197.84 |
| MLP | 800.15 | 1439.89 | 1236.88 | 1730.88 | 230.85 |
| | 431.97 | 1040.34 | 209.54 | 1108.32 | 483.36 |
| | 429.76 | 765.7 | 306.08 | 519.73 | 163.36 |
| | 426.27 | 117.05 | 228.26 | 88.28 | 232.8 |
| | 412.39 | 65.29 | 187.72 | 108.97 | 133.99 |

Table 12: Detail results of Robustness testing.

## 6.7 HYPERPARAMETERS FOR MULTI-TASK LEARNING

Table 13: Hyperparameter settings for multi-task learning.

| MLP, TreeNet and NerveNet | Value Tried |
|---|---|
| Network Shape | [64, 64] |
| Number of Iteration Per Update | 10 |
| Use KL Penalty | No |
| Learning Rate Scheduler | Adaptive |
| NerveNet | Value Tried |
| Number of Propogation Steps | 4 |
| NerveNet Variants | NerveNet-1 |
| Size of Nodes' Hidden State | 64 |
| Merge `joint` and `body` Node | Yes |
| Output Network | Separate |
| Disable Edge Type | Yes |
| Add Skip-connection from / to `root` | No |

## 6.8 SCHEMATIC FIGURES OF WALKERS AGENTS

We design `Walker` task-set, which contains five 2d walkers in the MuJoCo engine.

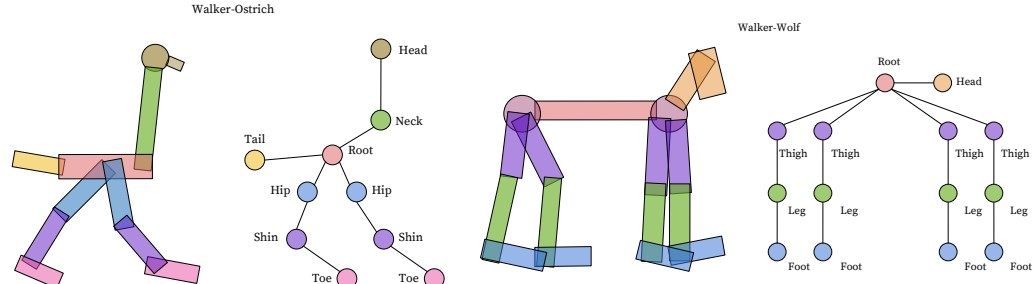

Figure 17: Schematic diagrams and auto-parsed graph structures of `Walker-Ostrich` and `Walker-Wolf`.

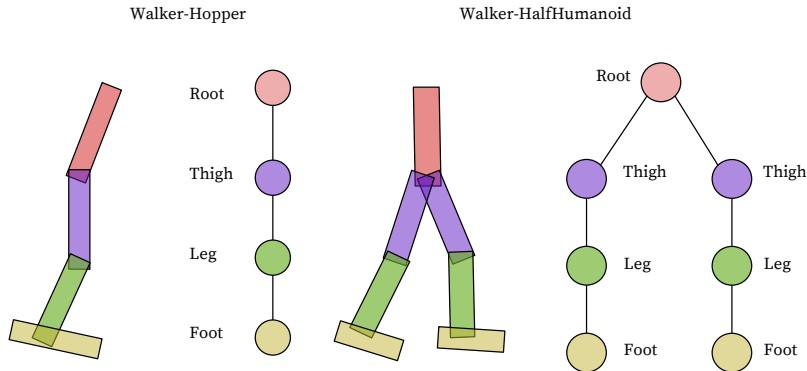

Figure 18: Schematic diagrams and auto-parsed graph structures of `Walker-Hopper` and `Walker-HalfHumanoid`.

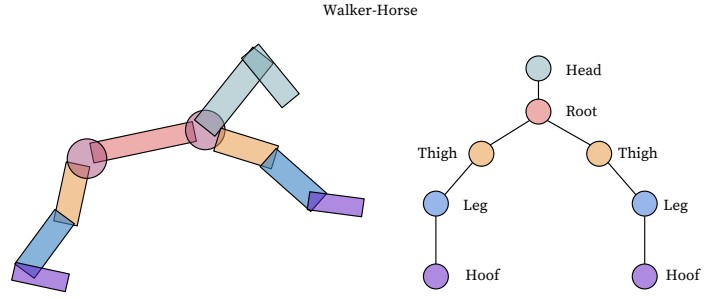

Figure 19: Schematic diagrams and auto-parsed graph structures of `Walker-Horse`.

