# OpenReview forum: "NerveNet: Learning Structured Policy with Graph Neural Networks"
_ICLR.cc/2018/Conference — Accept (Poster)_

### Official Review · AnonReviewer1 · 2017-11-25
**A nice paper that learns structured policy for control**

**Rating:** 7
**Confidence:** 3

**Review:**

This paper proposes NerveNet to represent and learn structured policy for continuous control tasks. Instead of using the widely adopted fully connected MLP, this paper uses Graph Neural Networks to learn a structured controller for various MuJoco environments. It shows that this structured controller can be easily transferred to different tasks or dramatically speed up the fine-tuning of transfer.

The idea to build structured policy is novel for continuous control tasks. It is an exciting direction since there are inherent structures that should be exploited in many control tasks, especially for locomotion. This paper explores this less-studied area and demonstrates promising results.

The presentation is mostly clear. Here are some questions and a list of minor suggestions:
1) In the Output Model section, I am not sure how the controller is shared. It first says that "Nodes with the same node type should share the instance of MLP", which means all the "joint" nodes should share the same controller. But later it says "Two LeftHip should have a shared controller." What about RightHip? or Ankle? They all belongs to the same node type "joint". Am I missing something here? It seems that in this paper, weights sharing is an essential part of the structured policy, it would be great if it can be described in more details.

2) In States Update of Propagation Model Section, it is not clear how the aggregated message is used in eq. (4).

3) Typo in Caption of Table 1: CentipedeFour not CentipedeSix.

4) If we just use MLP but share weights among joints (e.g. the weights from observation to action of all the LeftHips are constrained to be same), how would it compare to the method proposed in this paper?

In summary, I think that it is worthwhile to develop structured representation of policies for control tasks. It is analogue to use CNN that share weights between kernels for computer vision tasks. I believe that this paper could inspire many follow-up work. For this reason, I would recommend accepting this paper.

---

> ### Author Response · Authors · 2017-12-30
> **Response to AnonReviewer1**
>
> We thank the reviewer for the great suggestions regarding the quality of the paper, and we would like to bring your attention to the general comment above. We added new experiments in the latest revision.
>
> Q1: How does MLP with share weights among joints perform?
> A1: We name the variant proposed by the reviewer as MLP-Bind. Note that MLP-Bind and TreeNet are equivalent for the Snake agents, since the snakes only have one type of joint. We ran MLP-Bind for the zero-shot and fine-tuning experiments on centipedes. We summarize the results here:
>
> 1. Zero-shot performances of MLP-Bind and MLPAA are very similar. Both models have limited performance in the zero-shot scenario. Attached below is a sample table for several transfer tasks in centipedes (full results in the appendix of the revised draft)
>
> 2. For fine-tuning on ordinary centipedes from pretrained models, the performance is only slightly worse than when using MLP. In our experiment, in the two curves of transferring from CentipedeFour to CentipedeEight as well as CentipedeSix to CentipedeEight, MLP-Bind’s reward is 100-500 worse than MLPAA during fine-tuning.
>
> 3. For the Crippled agents, the MLP-Bind agent is stuck at around 800 reward. This might be due to MLP-Bind not being able to efficiently exploit the information of crippled and well-functioning legs.
>
> For the Average Reward:
> ----------------------------------------------------------------------------
>     Task     |    MLPAA   |   MLP-Bind   |    NerveNet
> ----------------------------------------------------------------------------
> 4to6          |    109.4      |      62.13      |    139.6
> 4to8          |    18.2        |      24.62      |    44.3
> ----------------------------------------------------------------------------
> 6to8          |    21.1        |      235.97    |    1674.9
> 6to10        |    -42.4       |      18.65      |    940.0
> ----------------------------------------------------------------------------
> 4toCp06    |    -5.1         |      11.47      |    47.6
> 4toCp08    |    5.1          |      7.34        |    40.0
> ----------------------------------------------------------------------------
> 6toCp08    |    36.5        |      29.09      |    523.6
> 6toCp10    |    12.8        |      8.32        |    504.0
> ----------------------------------------------------------------------------
>
> For the average distance the agents could run in one episode (see updated version for the details of average distance, which is another metrics to evaluate how well the agents perform.)
> ----------------------------------------------------------------------------
>     Task     |    MLPAA   |   MLP-Bind   |    NerveNet
> ----------------------------------------------------------------------------
> 4to6          |    545.3      |      62.13      |    577.3
> 4to8          |    62.0        |      24.62      |    146.9
> ----------------------------------------------------------------------------
> 6to8          |    87.8        |      235.97    |    10612.6
> 6to10        |    -17.0       |      18.65      |    6343.6
> ----------------------------------------------------------------------------
> 4toCp06    |    -22.5       |      11.47      |    91.1
> 4toCp08    |    -26.9       |      7.34        |    80.1
> ----------------------------------------------------------------------------
> 6toCp08    |    138.3       |      29.09      |    3117.3
> 6toCp10    |    13.6         |       8.32       |    3230.3
> ----------------------------------------------------------------------------
>
> The details of experiments we performed are updated in the appendix of the latest version.
>
>
> Q2: How the controller is shared? (“In the Output Model section, I am not sure how the controller is shared. It first says that "Nodes with the same node type should share the instance of MLP", which means all the "joint" nodes should share the same controller.”)
> A2: We clarified the Output Model section in the latest version.
> Nervnet:  nodes of the same type (joint, root, body) share the same state update function, e.g., GRU weights.
> Every node, regardless of its node type, shares the same output MLP instance.
>
>
> Q3: Typos and presentation regarding to Eq. (4).
> A3: We improved the clarity as per suggestions.

---

### Official Review · AnonReviewer2 · 2017-11-28
**Investigates an under-explored idea, but evaluation could be more compelling**

**Rating:** 6
**Confidence:** 3

**Review:**

The submission proposes incorporation of additional structure into reinforcement learning problems. In particular, the structure of the agent's morphology. The policy is represented as a graph neural network over the agent's morphology graph and message passing is used to update individual actions per joint.

The exposition is fairly clear and the method is well-motivated. I see no issues with the mathematical correctness of the claims made in the paper. However, the paper could benefit from being shorter by moving some details to the appendix (such as much of section 2.1 and PPO description).

Related work section could consider the following papers:

"Discrete Sequential Prediction of Continuous Actions for Deep RL"
Another approach that outputs actions per joint, although in a general manner that does not require morphology structure

"Generalized Biped Walking Control"
Considers the task of interactively changing limb lengths (your size transfer task) in a zero-shot manner, albeit with a non-neural network controller

The experimental results investigate the effects of various algorithm parameters, which is appreciated. However, a wider range of experiments would have been helpful to judge the usefulness of the proposed policy representation. In addition to robustness to limb length and disability perturbations, it would have been very nice to see multi-task learning that takes advantage of body structure (such as learning to reach for target with arms while walking with legs and being able to learn those independently, for example).

However, I do think using agent morphology is an under-explored idea and one that is general, since we tend to have access to this structure in continuous control tasks for the time being. As a result, I believe this submission would be of interest to ICLR community.

---

> ### Author Response · Authors · 2017-12-30
> **Response to AnonReviewer2**
>
> We thank the reviewer for the great suggestion regarding multi-task learning.
>
> Q1: “However, a wider range of experiments would have been helpful to judge the usefulness of the proposed policy representation … it would have been very nice to see multi-task learning that takes advantage of body structure”
> A1: We added the multi-task learning experiment. Please see the general comment above with additional experiments.
>
>
> Q2: Moving some details to the appendix.
> A2: We revised the paper and shortened the model’s section.
>
>
> Q3: Adding references.
> A3: Thanks for pointing out these two references. We included them in the latest version.
>
> We believe that the focus of “Discrete Sequential Prediction of Continuous Actions for Deep RL” paper (action space discretization) and the focus of our paper (using structure information) are different. Combining these ideas might further boost the performance of the agents.
>
> For the second paper, we agree that model-based control has been well studied, and this paper should be cited.

---

### Official Review · AnonReviewer3 · 2017-11-30
**Though occasionally unclear, the authors present an interesting approach to solving a well-scoped problem.**

**Rating:** 7
**Confidence:** 3

**Review:**

The authors present an interesting application of Graph Neural Networks to learning policies for controlling "centipede" robots of different lengths. They leverage the non-parametric nature of graph neural networks to show that their approach is capable of transferring policies to different robots more quickly than other approaches. The significance of this work is in its application of GNNs to a potentially practical problem in the robotics domain. The paper suffers from some clarity/presentation issues that will need to be improved. Ultimately, the contribution of this paper is rather specific, yet the authors show the clear advantage of their technique for improved performance and transfer learning on some agent types within this domain.

Some comments:
- Significant: A brief statement of the paper's "contributions" is also needed; it is unclear at first glance what portions of the work are the authors' own contributions versus prior work, particularly in the section describing the GNN theory.
- Abstract: I take issue with the phrase "are significantly better than policies learned by other models", since this is not universally true. While there is a clear benefit to their technique for the centipede and snake models, the performance on the other agents is mostly comparable, rather than "significantly better"; this should be reflected in the abstract.
- Figure 1 is instructive, but another figure is needed to better illustrate the algorithm (including how the state of the world is mapped to the graph state h, how these "message" are passed between nodes, and how the final graph states are used to develop a policy). This would greatly help clarity, particularly for those who have not seen GNNs before, and would make the paper more self-contained and easier to follow. The figure could also include some annotated examples of the input spaces of the different joints, etc. Relatedly, Sec. 2.2.2 is rather difficult to follow because of the lack of a figure or concrete example (an example might help the reader understand the procedure without having to develop an intuition for GNNs).
- There is almost certainly a typo in Eq. (4), since it does not contain the aggregated message \bar{m}_u^t.

Smaller issues / typos:
- Abstract: please spell out spell out multi-layer perceptrons (MLP).
- Sec 2.2: "servers" should be "serves"
- "performance By" on page 4 is missing a "."

Pros:
- The paper presents an interesting application of GNNs to the space of reinforcement learning and clearly show the benefits of their approach for the specific task of transfer learning.
- To the best of my knowledge, the paper presents an original result and presents a good-faith effort to compare to existing, alternative systems (showing that they outperform on the tasks of interest).

Cons:
- The contributions of the paper should be more clearly stated (see comment above).
- The section describing their approach is not "self contained" and is difficult for an unlearned reader to follow.
- The problem the authors have chosen to tackle is perhaps a bit "specific", since the performance of their approach is only really shown to exceed the performance on agents, like centipedes or snakes, which have this "modular" quality.

I certainly hope the authors improve the quality of the theory section; the poor presentation here brings down the rest of the paper, which is otherwise an easy read.

---

> ### Author Response · Authors · 2017-12-30
> **Response to AnonReviewer3**
>
> We thank the reviewer for the careful reading of our paper and suggestions.
>
> Q1: The problem the authors have chosen to tackle is perhaps a bit specific
> A1: Please see the general comment above with additional experiments.
>
>
> Q2: A brief statement of the paper's "contributions" is also needed.
> A2: We made the statement more clear in the latest version. Specifically, our main contribution is in exploring graph neural networks in reinforcement learning and investigating their ability to transfer structure. To the best of our knowledge, we are the first to address transfer learning for continuous control tasks.  We also make small contributions on the model side, i. e. GNNs. In particular, we introduce node type and associate an instance of an update function with each type. This fits the RL setting very well and also increases the model’s capacity.
>
>
> Q3: Abstract: I take issue with the phrase "are significantly better than policies learned by other models.
> A3: We agree and will modify the wording in the abstract. Our main claims were with respect to transferability, since our model has significant improvement in the zero-shot and transfer learning tasks.
>
>
> Q4: Another figure is needed to better illustrate the algorithm. Relatedly, Sec. 2.2.2 is rather difficult to follow because of the lack of a figure or concrete example
> A4: We added a new figure (Fig. 2 in the newest revision) to illustrate how the input state is constructed, how messages are passed between the nodes and how the final policy is being output.
>
>
> Q5: Typo in Eq. (4) and other minor issues.
> A5: Thanks for pointing this out. We corrected them in the latest version.

---

### Public Comment · ~George_Edward_Dahl1 · 2017-11-30
**Can you write your model as a message passing neural network?**

Is it possible to write your model as a message passing neural network, as in http://proceedings.mlr.press/v70/gilmer17a.html ? It looks closely related and readers might benefit from any explicit connections that can be made.

---

> ### Public Comment · (anonymous) · 2017-12-02
> **Message passing neural networks vs. graph neural networks**
>
> I think it is fair to frame their model as a graph neural network, as it closely resembles the "local transition function" proposed in the original graph neural network paper (Gori et al., 2009): http://ieeexplore.ieee.org/document/4700287/
>
> The only architectural difference that the authors propose here is to use a gated per-node update function right after the local transition function is evaluated - this mostly resembles the work from Li et al., 2015: https://arxiv.org/abs/1511.05493 (which is cited).

---

> ### Author Response · Authors · 2017-12-02
> **Connection with Message Passing Neural Network**
>
> Thanks for pointing out this paper!
> We do think we can rephrase our model as a message passing neural network (MPNNs) except some subtle differences, like the message function could take representations of both head and tail of an edge as input arguments in MPNNs whereas ours only takes representation of head to compute the message.
> We will add the discussion of these connections in the final version.

---

### Author Response · Authors · 2017-12-30
**Authors' General Response to Reviewers**

We thank all reviewers for their valuable comments and suggestions. We here address the common concerns/suggestions and summarize the modifications in the latest revision.

We first emphasize our contributions. In our work, we propose a model that exploits structure priors for continuous reinforcement learning. We show competitive performance on standard tasks, and focus on showing the model’s ability to perform transfer learning to different agents. To the best of our knowledge, we are the first to address transfer learning between different agents for continuous control tasks, whose even simplest sub-problems have been beyond the ability of the best models we have right now.


1. Multi-task Learning
One common concern among the reviewers is the lack of more diverse transfer experiments. We address this concern by performing an extensive set of multi-task experiments in our latest revision.  In particular, we trained one single network to control a broad range of diverse agents.

We create a Walker task-set which contains five 2d walkers. They have very different dynamics, from single legged hopper to two-legged ostrich with a tail and neck. Specifically, Walker-HalfHumanoid and Walker-Hopper are variants of Walker2d and Hopper, respectively, in the original MuJoCo Benchmarks. On the other hand, Walker-Horse (two-legged), Walker-Ostrich (two-legged), and Walker-Wolf (four-legged) are agents mimicking real animals. Just like real-animals, some of the agents have tails and a neck to help them to balance. The detailed schematic figures are in the appendix.

We refer to training  separate models of different weights for each agent as single-task learning, and sharing weights across multiple agents as multi-task learning. The results including our method and other competitors, e.g., MLP, are listed below:

TABLE 1
 Model                    | HalfHum| Hopper | Ostrich |   Wolf    |   Horse    | Average

    MLP    | Reward | 1775.75 | 1369.6 |  1198.9 | 1249.23 |  2084.1  |    /
    MLP    |   Ratio   |   57.7%   |  62.0% |  48.2%  |  54.5%    |  69.7%     | 58.6%

TreeNet  | Reward |   237.81 | 417.27 |  224.1   | 247.03   |  223.34    |    /
TreeNet  |   Ratio   |   79.3%   | 98.0%  |  57.4%  | 141.2%  |   99.2%    |  94.8%

NerveNet| Reward|  2536.52 | 2113.6 |  1714.6 | 2054.5   |  2343.6  |     /
NerveNet|  Ratio   |   96.3%   | 101.8% |  98.8%  | 105.9%  |   106.4%  |  101.8%

(Ratio indicates “the reward of multi-task” / “the reward of single-task baseline”)

From the results, we can see that our method significantly outperforms other models. The MLP models failed to learn a shared representation over the different tasks. Their performance drops significantly when shifting from single-task to multi-task learning, while the performance of the NerveNet remains the same. We also show training curves in the updated version of the paper.


2. Robustness
To  assess the model’s generalization ability, we added an experiment to evaluate how well the control policies can generalize from the training environment to slightly perturbed test environments, e.g. varying the mass or the torque of the walkers’ joints.

As pointed out by [1], the policy learned by MLP is very unstable and is typically overfit. Different from [1], where the authors improve the robustness via model ensembles, we show that NerveNet is able to improve robustness of the agent from the perspective of model’s structure, which means that NerveNet is able to improve robustness of the agent by exploiting priors and weight sharing in the model’s structure.
In this experiment, we perturbed the mass of the geometries (rigid bodies) in MuJoCo as well as the scale of the forces of the joints. We used the pretrained models with similar performance on the original task for both the MLP and NerveNet. We tested the performance in five agents from the “Walker” task set. The average performance is recorded in the figure below, and the specific details are summarized in the appendix of the latest paper revision.
Shown in the below table are the results of "performance of perturbed agents" / "training performance".

TABLE 2
     Model               | HalfHum  | Hopper |   Wolf    | Ostrich |  Horse   | Average

Mass |    MLP       |  33.28%    | 74.04%  | 94.68% | 59.23% | 40.61% | 60.37%
Mass | NerveNet |  95.87%   | 93.24%  | 90.13% |  80.2%  | 69.23% | 85.73%

STR    |     MLP      |   25.96%   | 21.77%  | 27.32% | 30.08%  | 19.80% | 24.99%
STR    | NerveNet |   31.11%   | 42.20%  | 42.84% | 31.41%  | 36.54% | 36.82%

In summary, we added (1) experiments on multi-task learning, (2) experiments on testing robustness, (3) included improved visualizations of the zero-shot learning experiments, and  added (4) more details in the appendix, e.g., hyper-parameters, the schematic figures of the “Walker” task-set agents.

[1] Towards Generalization and Simplicity in Continuous Control

---

### Decision · Program_Chairs · 2018-01-29
**ICLR 2018 Conference Acceptance Decision**

**Decision:**

Accept (Poster)

**Comment:**

An interesting application of graph neural networks to robotics. The body of a robot is represented as a graph, and the agent’s policy is defined using a graph neural network (GNNs/GCNs) over the graph structure.

The GNN-based policy network perform on par with best methods on traditional benchmarks, but shown to be very effective for transfer scenarios: changing robot size or disabling its components.  I believe that the reviewers' concern that the original experiments focused solely on centepedes and snakes were (at least partially) addressed in the author response: they showed that their GNN-based model outperforms MLPs on a dataset of 2D walkers.

Overall:
-- an interesting application
-- modeling robot morphology is an under-explored direction
-- the paper is  well written
-- experiments are sufficiently convincing (esp. after addressing the concerns re diversity and robustness).